EMBO
Molecular Medicine

# Enhancing protective microglial activities with a dual function TREM2 antibody to the stalk region

Kai Schlepckow[1,†], Kathryn M Monroe[2,†], Gernot Kleinberger[3,†,‡] (iD), Ludovico Cantuti-Castelvetri[1], Samira Parhizkar[3], Dan Xia[2], Michael Willem[3], Georg Werner[3], Nadine Pettkus[3], Bettina Brunner[1], Alice Sülzen[1], Brigitte Nuscher[3], Heike Hampel[3], Xianyuan Xiang[3,4], Regina Feederle[1,5,6] (iD), Sabina Tahirovic[1] (iD), Joshua I Park[2], Rachel Prorok[2], Cathal Mahon[2], Chun-Chi Liang[2], Ju Shi[2,§], Do Jin Kim[2], Hanna Sabelström[2], Fen Huang[2], Gilbert Di Paolo[2], Mikael Simons[1,5,7], Joseph W Lewcock[2,*] (iD) & Christian Haass[1,3,5,**] (iD)

## Abstract

Triggering receptor expressed on myeloid cells 2 (TREM2) is essential for the transition of homeostatic microglia to a disease-associated microglial state. To enhance TREM2 activity, we sought to selectively increase the full-length protein on the cell surface via reducing its proteolytic shedding by A Disintegrin And Metalloproteinase (i.e., α-secretase) 10/17. We screened a panel of monoclonal antibodies against TREM2, with the aim to selectively compete for α-secretase-mediated shedding. Monoclonal antibody 4D9, which has a stalk region epitope close to the cleavage site, demonstrated dual mechanisms of action by stabilizing TREM2 on the cell surface and reducing its shedding, and concomitantly activating phospho-SYK signaling. 4D9 stimulated survival of macrophages and increased microglial uptake of myelin debris and amyloid β-peptide *in vitro*. *In vivo* target engagement was demonstrated in cerebrospinal fluid, where nearly all soluble TREM2 was 4D9-bound. Moreover, in a mouse model for Alzheimer's disease-related pathology, 4D9 reduced amyloidogenesis, enhanced microglial TREM2 expression, and reduced a homeostatic marker, suggesting a protective function by driving microglia toward a disease-associated state.

**Keywords** Alzheimer's disease; amyloid β-peptide; microglia; therapeutic antibody; TREM2

**Subject Categories** Immunology; Neuroscience; Pharmacology & Drug Discovery

## Introduction

Alzheimer's disease (AD) and related disorders are devastating diseases threatening our aging society, and still, no cure is available. Since a number of recent clinical trials using amyloid β-peptide (Aβ)-based therapeutic approaches failed to improve cognition or even worsened the clinical outcome of patients (Egan *et al*, 2019), novel targets are desperately needed. Importantly, it appears that anti-Aβ therapeutics may have to be administered decades before symptom onset, since at the time patients are enrolled in clinical studies they are amyloid positron emission tomography (PET)-positive (Sevigny *et al*, 2016) even if they have not yet developed overt dementia. We therefore need to develop strategies to interfere with the amyloid cascade immediately after it has been initially triggered by Aβ and may become independent of its further *de novo* production.

In addition to the selective deposition of amyloidogenic proteins, neuroinflammation associated with microgliosis is a common feature of many neurodegenerative disorders (Ransohoff, 2016). Recent genome-wide association studies strongly substantiated a central role of innate immunity for neurodegeneration by identifying a number of risk variants in genes that are exclusively expressed within microglia in the brain. Among them, coding variants in the

1  German Center for Neurodegenerative Diseases (DZNE) Munich, Munich, Germany
2  Denali Therapeutics Inc., South San Francisco, CA, USA
3  Metabolic Biochemistry, Biomedical Center (BMC), Faculty of Medicine, Ludwig-Maximilians-Universität München, Munich, Germany
4  Graduate School of Systemic Neuroscience, Ludwig-Maximilians-Universität München, Munich, Germany
5  Munich Cluster for Systems Neurology (SyNergy), Munich, Germany
6  Helmholtz Center Munich, German Research Center for Environmental Health, Institute for Diabetes and Obesity, Core Facility Monoclonal Antibody Development, Munich, Germany
7  Institute of Neuronal Cell Biology (TUM-NZB), Munich, Germany
   *Correspondence: Tel: +1 (650) 745-5247; E-mail: lewcock@dnli.com
   **Correspondence: Tel: +49 (0) 89-4400-46549; E-mail: christian.haass@dzne.de
   †These authors contributed equally to this work
   ‡Present address: ISAR Bioscience GmbH, Planegg, Germany
   §Present address: Jazz Pharmaceuticals, Palo Alto, CA, USA

triggering receptor expressed on myeloid cells 2 (TREM2) increase the risk for late-onset AD as much as the apolipoprotein ε4 allele (Guerreiro et al, 2013; Jonsson & Stefansson, 2013; Jonsson et al, 2013; Rayaprolu et al, 2013; Borroni et al, 2014; Cuyvers et al, 2014). The disease-associated variants appear to cause a loss of function of TREM2 through a variety of mechanisms, including inhibition of cellular maturation of TREM2, destroying essential lipid binding sites within the immunoglobulin region of the ectodomain, and enhancing its shedding (Kleinberger et al, 2014, 2017; Mazaheri et al, 2017; Schlepckow et al, 2017; Ulland et al, 2017; Song et al, 2018). Full-length TREM2, in association with DAP12, forms a heteromeric complex required to activate phospho-SYK signaling (Colonna, 2003). Signaling appears to be terminated by α-secretase-mediated shedding of the TREM2 ectodomain (Fig 1A) (Wunderlich et al, 2013; Kleinberger et al, 2014), although very recent data suggest that soluble TREM2 (sTREM2) may also have signaling functions (Zhong et al, 2017, 2019), and requires further investigation. Loss of TREM2-mediated signaling locks microglia in a homeostatic state and inhibits their transition to disease-associated microglia (DAM) (Krasemann et al, 2017; Mazaheri et al, 2017), which are phenotypically characterized by enhanced migration, chemotaxis, and phagocytosis (Keren-Shaul et al, 2017; Mazaheri et al, 2017). Furthermore, lack of functional TREM2 appears to be associated with reduced cellular proliferation and death of phagocytes as well as reduced energy metabolism and cerebral blood flow in mice (Ulland et al, 2015, 2017; Kleinberger et al, 2017). TREM2 has been shown to be involved in lipid sensing, a function which is diminished by the AD-associated TREM2 R47H variant (Wang et al, 2015). Furthermore, loss of TREM2 results in downregulation of genes involved in lipid metabolism (Keren-Shaul et al, 2017; Griciuc et al, 2019). Consistent with these observations, TREM2 appears to be required for elimination of damaged myelin by microglia (Cantoni et al, 2015). Moreover, TREM2-deficient microglia were shown to be overloaded with cholesterol esters (CEs) as well as

other lipid species as a result of chronic phagocytic challenge of myelin debris induced by a cuprizone diet (Nugent et al, 2019), indicating that TREM2 is important for lipid processing in addition to phagocytosis.

Quantitative analyses of the cleaved sTREM2 ectodomain in the cerebrospinal fluid of patients with sporadic and autosomal dominant AD (ADAD) revealed enhanced levels at the mild cognitive impairment stage (Suarez-Calvet et al, 2016b, 2019) and up to 5 years before the estimated onset of ADAD (Suarez-Calvet et al, 2016a), respectively. Moreover, sTREM2 positively correlates with total tau and phospho-tau but not with Aβ (Suarez-Calvet et al, 2016b). Thus, CSF sTREM2 seems to be associated with tau pathology. Moreover, microglial activation and associated cellular functions are protective for cognition in a mouse model of AD (Focke et al, 2019), and ameliorate retinal degeneration (O'Koren et al, 2019). Furthermore, high levels of CSF sTREM2 predict a better outcome of cognition and reduced hippocampal shrinkage in the Alzheimer's disease neuroimaging initiative cohort (Ewers et al, 2019). Taken in combination with the observation that TREM2 loss of function increases AD risk, this suggests that enhancing TREM2 signaling represents a therapeutic strategy that is downstream of the initial Aβ-mediated trigger of the amyloid cascade.

Since α-secretase-mediated shedding of TREM2 (Kleinberger et al, 2014; Schlepckow et al, 2017; Thornton et al, 2017) is sufficient to abrogate cell-autonomous signaling, we hypothesized that elevating TREM2 levels on the cell surface would enhance potentially disease-modulating microglial functions. Indeed, inhibition of α-secretase activity with a protease inhibitor enhanced the phagocytic capacity of microglial-like BV2 cells (Kleinberger et al, 2014). However, α-secretase competes with the amyloidogenic pathway and limits Aβ production (Haass & Selkoe, 1993); therefore, its inhibition in AD would be not be efficacious. Moreover, a disintegrin and metalloproteinase (ADAM, i.e., α-secretase) 10 and ADAM17 cleave numerous substrates within the brain, many of which

**Figure 1. Screening and molecular characterization of anti-mouse TREM2 antibodies.**

A  Schematic representation of TREM2 processing by ADAM10/17. Cleavage occurs C-terminal of residue His 157. The entire ectodomain (residues 19–171) was used for immunization of rats to generate TREM2 antibodies. CTF, C-terminal fragment; sTREM2, soluble TREM2.

B  Immunoblot analysis of membrane fractions of HEK293 Flp-In cells stably overexpressing both mouse TREM2 and mouse DAP12 upon treatment with 4D9 antibody reveals increased levels of membrane-bound TREM2 similar to what can be achieved by ADAM protease inhibition using the GM6001 inhibitor. An isotype antibody was used as a negative control. Calnexin served as a loading control. Levels of membrane-bound TREM2 were quantified by MSD ELISA. Data represent the mean $\pm$ SEM ($n$ = 3). One-way ANOVA, Tukey's *post hoc* test; $P$ (DMSO vs GM) = 0.0011; $P$ (DMSO vs isotype) = 0.992; $P$ (isotype vs 4D9) = 0.0005; n.s., not significant.

C  Immunoblot analysis of conditioned media from HEK293 Flp-In cells stably overexpressing both mouse TREM2 and mouse DAP12 upon treatment with 4D9 antibody reveals decreased levels of sTREM2 similar to what can be achieved by ADAM protease inhibition using the GM6001 inhibitor. An isotype antibody was used as a negative control. sAPPα served as a loading control. Note that heavy and light chains of the antibodies used for treatment are also detected and annotated. Levels of sTREM2 were quantified by MSD ELISA. Data represent the mean $\pm$ SEM ($n$ = 3). One-way ANOVA, Tukey's *post hoc* test; $P$ (DMSO vs GM) < 0.0001; $P$ (DMSO vs isotype) = 0.6372; $P$ (isotype vs 4D9) < 0.0001; n.s., not significant.

D  4D9 antibody selectively detects TREM2 on the cell surface of HEK293 Flp-In cells stably overexpressing mouse TREM2 and mouse DAP12. An anti-HA antibody was used as a positive control, while empty vector-transfected HEK293 Flp-In cells were used as a negative control. Scale bar = 10 μm.

E  Peptide ELISAs detect anti-mouse TREM2 antibody binding to tiled stalk region peptides, full-length stalk peptide, or a truncated ADAM cleavage site peptide. The binding epitope of 4D9 antibody is located 12-amino acids N-terminal of the ADAM cleavage site at His 157.

F  Sequence comparison of mouse TREM2 and human TREM2 shows substantial sequence conservation around the 4D9 epitope (upper panel). Immunoblot analysis demonstrates that antibody 4D9 is highly specific for mouse TREM2 and does not detect human TREM2 or mouse TREM1 (lower panel).

G  4D9 binding to the mouse TREM2 ECD is competed off by a stalk region peptide. A competition ELISA demonstrates that a dose titration of stalk peptide reduces binding of 4D9 to TREM2 ECD with an EC50 of 1.3 μM. Data represent the mean $\pm$ SEM ($n$ = 3). ECD, extracellular domain.

H  Surface plasmon resonance binding kinetics of increasing concentrations of 4D9 antibody to mouse TREM2 ECD evaluated by Biacore, $k_{on}$ = 5.9 × 10$^5$ M$^{-1}$ s$^{-1}$, $k_{off}$ = 4.0 × 10$^{-5}$ s$^{-1}$, $K_D$ = 68 pM. 4D9 binding to human TREM2 or mouse TREM1 was undetectable.

Data information: Statistical evaluations are displayed as follows: **$P$ < 0.01; ***$P$ < 0.001; ****$P$ < 0.0001.
Source data are available online for this figure.

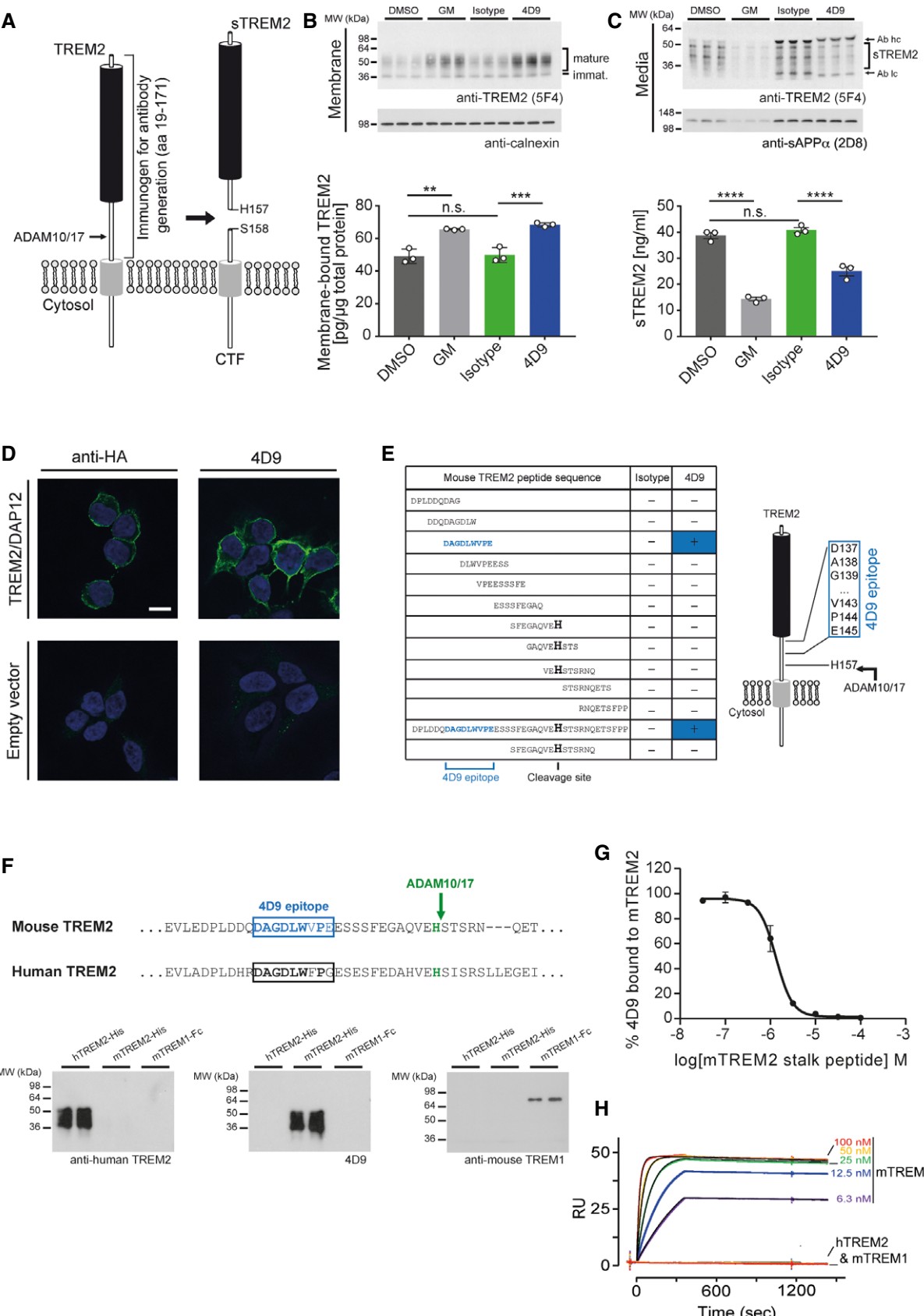

Figure 1.

stimulate essential physiological signaling pathways (Kuhn *et al*, 2016). We therefore sought to selectively inhibit TREM2 cleavage without affecting shedding of other ADAM10/17 substrates. We and others have previously identified the cleavage site of TREM2 after His 157 within the stalk region of TREM2 (Feuerbach *et al*, 2017; Schlepckow *et al*, 2017; Thornton *et al*, 2017) (Fig 1A). In this study, we screened for monoclonal antibodies binding to the stalk region encompassing the cleavage site with the purpose of stabilizing membrane-bound TREM2 and selectively enhancing TREM2-dependent protective functions in microglia.

## Results

### Screening and molecular characterization of anti-TREM2 antibodies

To identify antibodies capable of increasing TREM2 on the cell surface, we raised monoclonal antibodies to the recombinantly expressed ecto-domain of mouse TREM2 (amino acids 19–171; Fig 1A). Among the antibodies tested, 4D9, a rat IgG2a antibody, was selected for additional studies based on its ability to significantly enhance cell membrane TREM2 levels and to reduce sTREM2 levels similar to an ADAM protease inhibitor (Fig 1B and C). Immunohistochemistry showed that the 4D9 antibody bound to the cell surface of mouse TREM2-expressing HEK293 cells, while empty vector-expressing control cells did not show staining (Fig 1D). Antigen mapping using tiling peptides along the stalk region of TREM2 revealed specific binding of 4D9 to N-DAGDLWVPE, a 9-amino acid peptide N-terminal to the ADAM cleavage site (Fig 1E). Although this epitope shows substantial sequence similarities to human TREM2 (Fig 1F), 4D9 did not detect recombinant human TREM2 by Western blotting (Fig 1F). Furthermore, consistent with a lack of sequence conservation of the 9-amino acid region between mouse TREM1 and TREM2, 4D9 binding to mouse TREM1 was not detected (Fig 1F), demonstrating that 4D9 is indeed selective for mouse TREM2. To further confirm specific binding of 4D9 to the stalk region epitope, peptide competition experiments were performed. This demonstrated efficient competition of a TREM2 stalk peptide for 4D9 binding to the extracellular protein (Fig 1G).

Antibody–antigen affinity was determined by surface plasmon resonance (SPR) demonstrating a monovalent $K_D$ of 68pM for mouse TREM2 (Fig 1H). Consistent with the Western blot data (see Fig 1F), human TREM2 and mouse TREM1 did not result in any SPR response (Fig 1H).

High-affinity cell-surface TREM2 binding was established for 4D9 using HEK293 cells stably expressing TREM2. A dose response demonstrated a cell binding $EC_{50}$ of 0.29 nM, while an isotype control antibody did not show measurable binding (Fig 2A). Next, we generated Fab fragments of 4D9, which specifically bound cell-surface TREM2 on HEK293 cells with an EC50 of 0.17 nM (Fig 2A). *In vitro* peptide cleavage assays using recombinant ADAM17 revealed that the full-length 4D9 antibody, as well as 4D9 Fab, significantly blocked cleavage of a TREM2 stalk peptide substrate (Fig 2B). Thus, 4D9 appears to sterically hinder access of ADAM17 to its substrate. However, in a cell-based assay, only full-length IgG 4D9 antibody, but not 4D9 Fab, potently reduced shedding of TREM2 in a dose-dependent manner with an EC50 of 2.3 nM (Fig 2C). Given that 4D9 reduced shedding and enhanced cell-surface levels of full-length TREM2, we next evaluated the effects on downstream signaling. We therefore investigated p-SYK activity in the presence or absence of 4D9 and an isotype control in HEK293 cells expressing mouse TREM2 and its signaling adapter DAP12. This revealed a dose-dependent increase in p-SYK upon addition of 4D9 but not 4D9 Fab to the culture media of the cells (Fig 2D). Furthermore, anionic liposome ligand (Shirotani *et al*, 2019) stimulated TREM2 signaling 2.3-fold by full-length 4D9 but not by 4D9 Fab or the isotype control (Fig 2E). These data therefore suggest that the monoclonal antibody 4D9 enhances TREM2-dependent signaling via bivalent binding which cross-links TREM2 on the plasma membrane. Bivalent TREM2 binding is required for both signaling and shedding blocking mechanisms of TREM2 antibody function, potentially as a result of the inability of proteases to accept dimeric substrates (Fig 2F).

### 4D9 antibody modulates TREM2 function in primary macrophages and microglia

We next investigated whether TREM2-dependent functions of myeloid cells can be modulated by 4D9. For these studies, 4D9

**Figure 2. 4D9 antibody stimulates TREM2-dependent SYK signaling and blocks ADAM17-mediated TREM2 shedding.**

A   Flow cytometry dose–response curve for cell binding of 4D9 mAb (EC50 = 0.29 nM), 4D9 Fab (EC50 = 0.17 nM), and isotype to HEK cells stably overexpressing mouse TREM2. Data represent the mean ± SEM (*n* = 2).

B   *In vitro* ADAM17 sheddase activity is blocked by 4D9-effectorless mAb and 4D9 Fab fragment but not an isotype control. Fluorescence polarization of FAM-conjugated TREM2 stalk peptide was detected in the presence or absence of ADAM17 and 4D9 mAb, 4D9 Fab, and isotype control. Data represent the mean ± SEM (*n* = 6). One-way ANOVA, Tukey's *post hoc* test; P (4D9 Fab vs 4D9 mAb) = 0.8855; P (4D9 Fab vs uncleaved) < 0.0001; P (4D9 mAb vs uncleaved) < 0.0001; n.s., not significant.

C   ELISA-mediated quantification of sTREM2 in conditioned media from HEK293 cells stably overexpressing mouse TREM2 treated with a dose titration of 4D9 mAb (EC50 = 2.3 nM), 4D9 Fab, or isotype for 18 h. Data represent the mean ± SEM (*n* = 3).

D   AlphaLISA-mediated quantification of p-SYK levels in HEK293 Flp-In cells stably overexpressing mouse TREM2 and mouse DAP12 treated with a dose titration of 4D9 mAb, 4D9 Fab, or isotype for 5 min. Data represent the mean ± SEM (*n* = 3).

E   AlphaLISA-mediated quantification of p-SYK levels in HEK293 Flp-In cells stably overexpressing mouse TREM2 and mouse DAP12 or empty vector treated with 1 mg/ml POPC/POPS liposomes and 20 μg/ml 4D9 mAb, 4D9 Fab, or isotype for 5 min. p-SYK levels were also determined for cells treated with liposomes only. Data represent the mean ± SEM (*n* = 3). Two-way ANOVA, Tukey's *post hoc* test (cell line effect: $F_{1,16}$ = 365.7, $P \leq 0.0001$; treatment effect: $F_{3,16}$ = 39.35, $P < 0.0001$; cell line × treatment effect: $F_{3,16}$ = 38.75, $P > 0.0001$); P (no Ab vs isotype) = 0.6218; P (no Ab vs 4D9 mAb) < 0.0001; P (no Ab vs 4D9 Fab) = 0.7301; P (isotype vs 4D9 mAb) < 0.0001; P (isotype vs 4D9 Fab) > 0.9999; P (4D9 mAb vs 4D9 Fab) < 0.0001; n.s., not significant.

F   Schematic representation of the proposed mechanism of action of antibody 4D9. Binding of 4D9 to TREM2 leads to receptor clustering on the cell surface, thereby driving downstream p-SYK signaling. At the same time, cell-surface levels are enhanced by inhibition of ectodomain shedding, potentially because of the inability of proteases to cleave dimeric substrates.

Data information: Statistical evaluations are displayed as follows: ***P < 0.001; ****P < 0.0001.

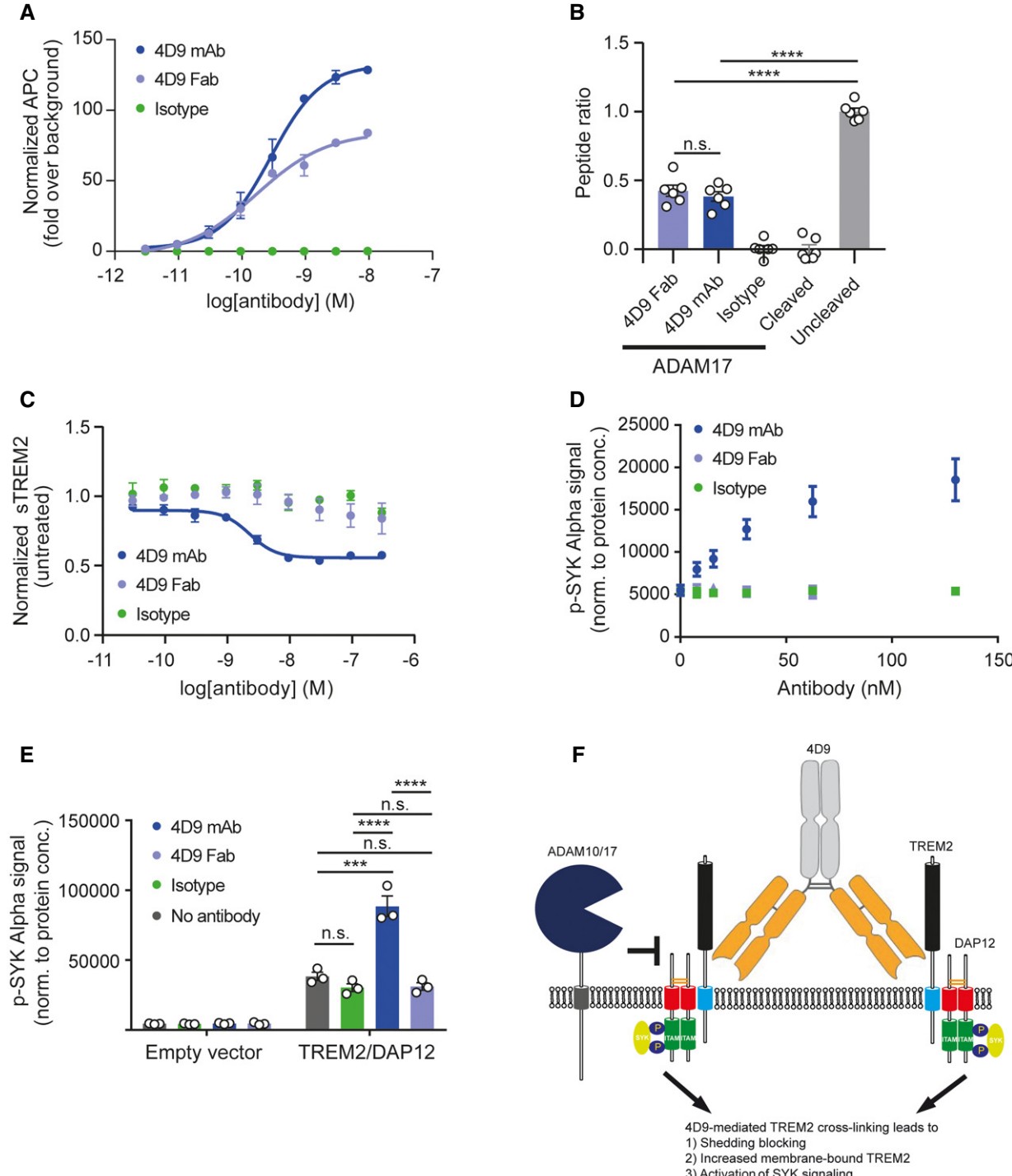

**Figure 2.**

was sequenced and reformatted onto an effectorless human IgG1 backbone carrying mutations that abolish effector function to avoid confounds of off-target impact of FcR interactions on primary myeloid cells (for details, see Materials and Methods). To demonstrate that 4D9 binds endogenous cell-surface TREM2 in myeloid cells, bone marrow-derived macrophages (BMDMs) were incubated with biotinylated 4D9 or isotype control antibody. Cell-surface binding of biotinylated 4D9 was confirmed by FACS with

fluorophore-conjugated streptavidin detection and found to be statistically significant on BMDM differentiated from three wild-type mice compared with an isotype control (Fig 3A). Binding was not detected on BMDM from three TREM2 KO mice, confirming the specificity of this interaction (Fig 3A). We then examined the impact of 4D9 on TREM2 shedding by Western blotting of membrane fractions isolated from six independent BMDM preparations. This demonstrated that 4D9 treatment increased total

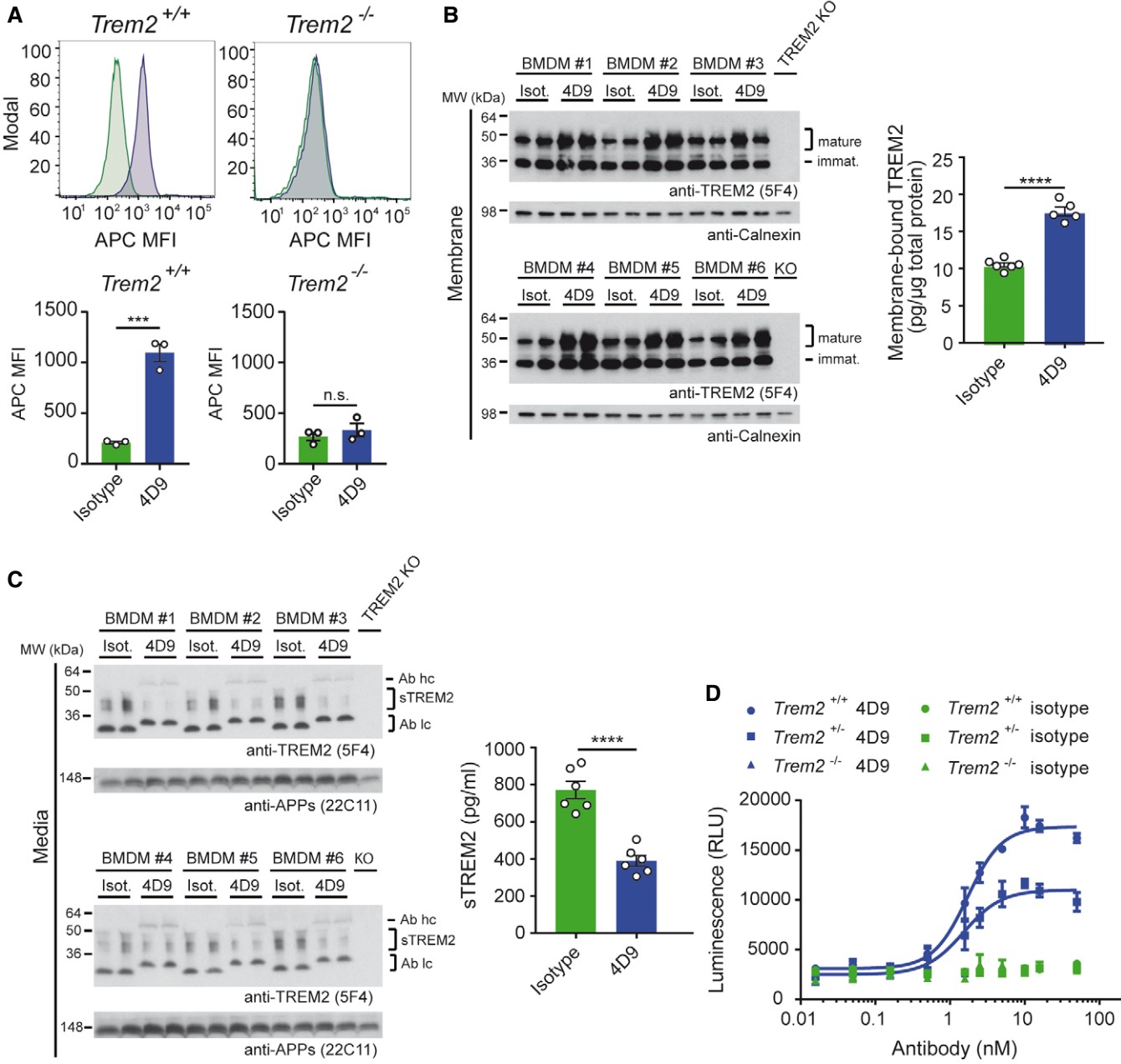

**Figure 3. 4D9 antibody modulates TREM2 expression in macrophages and stimulates survival.**

A   Flow cytometry detection of cell-surface binding of 4D9 (blue) and isotype (green) to wild-type and *Trem2*$^{-/-}$ mouse bone marrow-derived macrophages. Representative histograms are shown as MFI (mean fluorescent intensity). Graphs represent the median signal intensity ± SEM (*n* = 3). Student's *t*-test (two-tailed); *Trem2*$^{+/+}$: *P* = 0.0005; *Trem2*$^{-/-}$: *P* = 0.4435; n.s., not significant.

B   Immunoblot analysis of membrane fractions of bone marrow-derived macrophages (BMDMs) upon treatment with 4D9 antibody reveals increased levels of membrane-bound TREM2 relative to an isotype antibody. BMDMs from TREM2 knockout mice were included to show the specificity of the anti-TREM2 antibody. Calnexin served as a loading control. Levels of membrane-bound TREM2 were quantified by MSD ELISA. TREM2 antibody clone 5F4 was used for detection. Data represent the mean ± SEM (*n* = 5–6). Unpaired *t*-test (two-tailed) with Welch's correction; *P* < 0.0001. No gender-specific differences could be observed.

C   Immunoblot analysis of sTREM2 in conditioned media from BMDM upon treatment with 4D9 and isotype antibodies. BMDMs from TREM2 knockout mice were included to show the specificity of the anti-TREM2 antibody. Soluble APP served as a loading control. Note that heavy and light chains of the antibodies used for treatment are also detected and annotated as follows: Ab hc, antibody heavy chain; Ab lc, antibody light chain. Levels of sTREM2 were quantified by MSD ELISA. TREM2 antibody clone 5F4 was used for detection. Data represent the mean ± SEM (*n* = 6). Unpaired *t*-test (two-tailed) with Welch's correction; *P* < 0.0001. No gender-specific differences could be observed.

D   Wild-type, *Trem2*$^{+/-}$, and *Trem2*$^{-/-}$ BMDMs were plated in low concentration of M-CSF on 4D9 or isotype-coated wells for 5 days, and cellular ATP levels were measured by luminescence detection to indicate cell viability. 4D9 and isotype data for each genotype represent the mean ± SEM (*n* = 2 and *n* = 1, respectively).

Data information: Statistical evaluations are displayed as follows: ***P* < 0.001; *****P* < 0.0001.
Source data are available online for this figure.

full-length TREM2 (Fig 3B) and reduced levels of sTREM2 in conditioned media (Fig 3C). Thus, 4D9 reduces shedding and increases TREM2 on the cell surface of BMDMs.

To study functional consequences of 4D9 treatment, we investigated survival of BMDMs after M-CSF withdrawal. BMDMs were cultured in low M-CSF concentrations in the media on plates coated with increasing amounts of 4D9 for 5 days, and the total number of cells present was detected by total ATP quantification. This revealed that 4D9 increased cell survival of BMDMs in a potent and dose-dependent manner with an EC50 of 1.7 nM, whereas the control antibody failed to rescue survival defects induced by growth factor withdrawal (Fig 3D). Next, we asked whether a partial loss of function of TREM2 can also be rescued by 4D9. To do so, we treated BMDMs derived from heterozygous TREM2 knockout animals. This revealed that 4D9 significantly increased survival of BMDMs after a partial loss of function of TREM2 with an EC50 of 1.5 nM although not to the extent observed in TREM2 wild-type cells (Fig 3D). Taken together, these findings demonstrate that the 4D9 antibody enhances macrophage survival and increases cell-surface TREM2 via a dual mechanism of action: reduction in shedding by ADAM10/17 and enhancement of receptor signaling.

We next investigated whether 4D9 enhances TREM2-dependent functions in microglia. First, we tested whether 4D9 specifically detects TREM2 on the surface of microglia. Indeed, 4D9 selectively binds to the cell surface of primary microglia from wild-type mice, whereas no binding is observed to microglia derived from a $Trem2^{-/-}$ (Fig 4A). Moreover, addition of antibody 4D9 to the media of primary microglia significantly increased the percentage of both myelin (Fig 4B and C) and Aβ42 (Fig 4B and D)-positive cells, while *Escherichia coli* uptake remained unaltered with 4D9 treatment (Fig 4B and E). Thus, 4D9 enhances TREM2-dependent functions in myeloid phagocytes of the periphery and the central nervous system.

### Pharmacokinetics and *in vivo* evidence of 4D9 target engagement

We first evaluated the pharmacokinetics of the 4D9-effectorless antibody compared with an isotype control in wild-type mice. Plasma levels of human IgG were evaluated at 1 and 24 h, and 4 and 7 days after 10 mg/kg intravenous tail injection. 4D9 demonstrated peripheral clearance equivalent to that of a normal IgG (Fig 5A), and no evidence for an immune response to the humanized 4D9 was observed on this timescale. As expected, brain antibody 4D9 concentrations were much lower than plasma (Fig 5B).

To provide *in vivo* evidence that levels of antibody in brain were sufficient for TREM2 target engagement (TE), we determined the amount of 4D9-bound sTREM2 relative to total sTREM2 present in the cerebrospinal fluid (CSF) from dosed wild-type mice at various time points post-dose (Fig 5C). We developed a CSF-based antibody bound: total sTREM2 assay as it is a clinically translatable approach to demonstrate TREM2 TE with a relevant biofluid, and can be connected to further pre-clinical evidence of brain microglial response. We found that a single injection of 50 mg/kg led to almost complete saturation of sTREM2 with antibody 4D9 after 24 h, which gradually reduced to 50% 10 days following a single dose of 4D9 (Fig 5D). In a second study, analysis of the fraction of total sTREM2 bound to 4D9 in CSF of wild-type mice (Fig 5C) after antibody doses ranging from 0.1 µg/kg to 50 mg/kg showed that full engagement

of CSF sTREM2 can be achieved 24 h after a single dose of 10 mg/kg (Fig 5E). Taken together, these findings indicate robust TE of 4D9 *in vivo*.

### 4D9 antibody modulates TREM2 functions within the brain

To provide direct evidence that 4D9 modulates TREM2 and its functions within the brain, we measured the total level of TREM2 in the brain of wild-type mice 24 h after injecting 10, 50, or 100 mg/kg of 4D9 antibody. This revealed that 4D9 increased total brain TREM2 in a dose-dependent manner (Fig 5F), suggesting that antibodies with mechanisms similar to 4D9 could be used to modulate TREM2 functions *in vivo*. To determine the impact of 4D9 in a disease-relevant model, 6-month-old APP knock-in mice (Saito *et al*, 2014) and wild-type mice were treated with 50 mg/kg of antibody 4D9 or an isotype control in three-day intervals over a 10-day duration (Fig 6A). No difference in TREM2 expression of wt mice was observed upon 4D9 treatment, which may be due to the lower sensitivity of immunohistochemistry as compared to quantitative ELISA determinations shown in Fig 5F. Control-treated APP knock-in mice showed elevated TREM2 expression as compared to non-transgenic C57BL6 mice (Fig 6B and C), reflecting a shift to a disease-associated state (Krasemann *et al*, 2017). Strikingly, treatment with antibody 4D9 resulted in further elevation of microglial TREM2 expression (Fig 6B and C), whereas the total number of microglia remained unchanged (Fig 6D). In contrast, the total number of cells positive for P2RY12, a marker for homeostatic microglia (Butovsky *et al*, 2014; Krasemann *et al*, 2017; Mazaheri *et al*, 2017), was selectively decreased upon treatment with antibody 4D9 as compared to the isotype control (Fig 6E and F). These findings therefore suggest that 4D9 promotes the transition from homeostatic microglia to DAM. Moreover, increased TREM2 and decreased P2RY12 expression was specifically observed in areas where microglia apparently clustered around amyloid plaques (Fig 6B and E). Consistent with this observation, we observed a reduced amyloid plaque load throughout the cortex upon treatment with 4D9 (Fig 7A and B). Higher magnification revealed that 4D9 treatment predominantly caused a reduction in the fuzzy plaque halo (Fig 7C). Quantification demonstrated that 4D9 treatment significantly reduced amyloid plaque area and coverage within the cortex (Fig 7D and E). This was further confirmed by Western blot analysis of soluble and insoluble Aβ demonstrating that 4D9 treatment specifically results in a reduction in soluble but not insoluble Aβ (Fig 7F). Furthermore, ELISA-mediated quantification of soluble and insoluble brain fractions revealed a selective reduction in soluble Aβ42 but not insoluble Aβ42 (Fig 7G and H). Thus, our findings indicate that antibody 4D9 modulates beneficial microglial functions *in vitro* and *in vivo*.

## Discussion

Neurodegenerative diseases are currently untreatable. We therefore sought a new therapeutic strategy, with the goal to modulate microglial function as a disease-modifying mechanism. Microgliosis is observed in numerous neurological disorders, and only recently evidence has been presented that TREM2-dependent microglial functions appear to be protective in AD models (Ulland *et al*, 2017; Focke *et al*, 2019; Parhizkar *et al*, 2019), AD patients (Ewers *et al*,

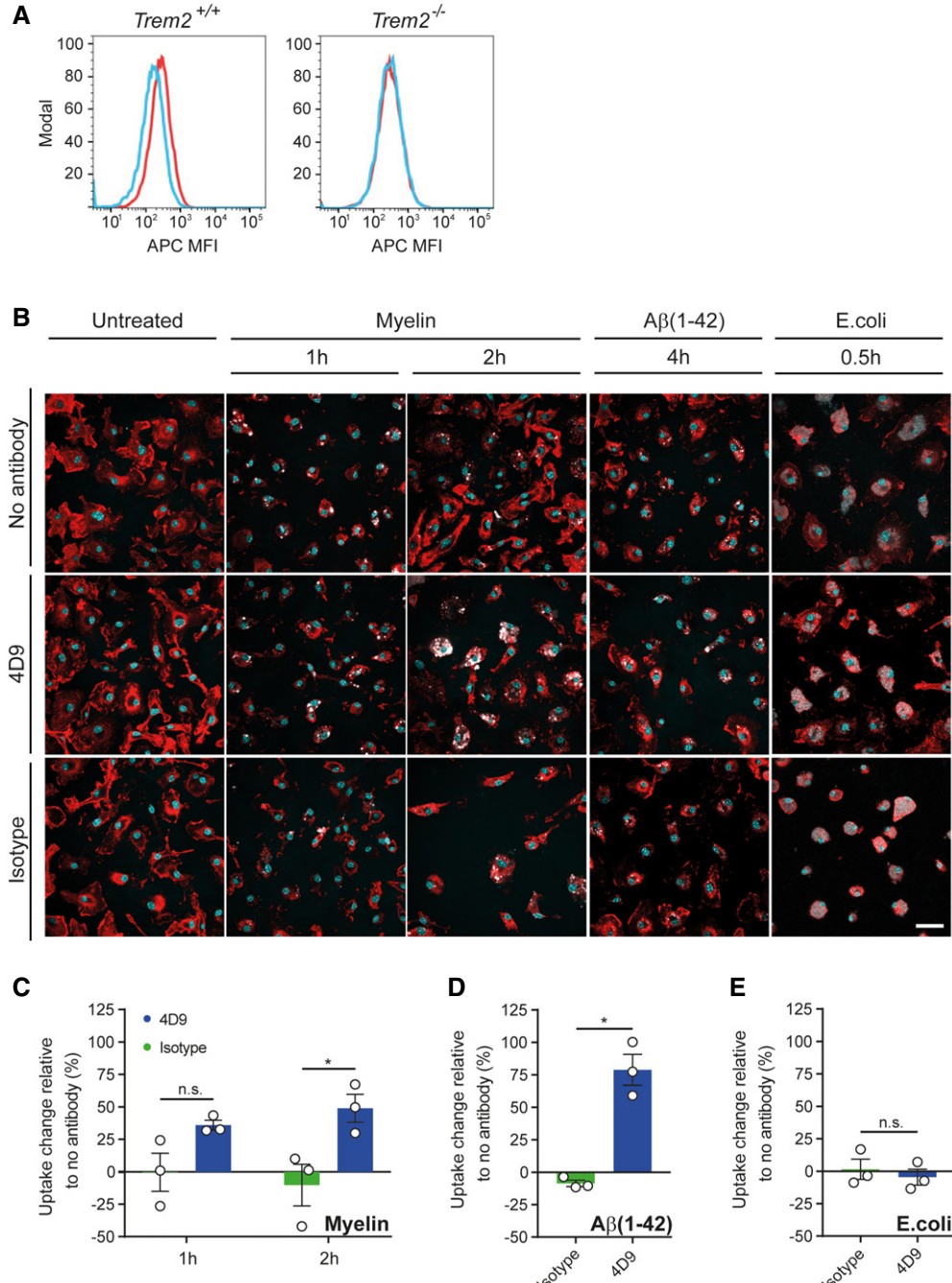

**Figure 4. 4D9 antibody promotes myelin and Aβ(1–42) phagocytosis in primary microglia.**

A   Flow cytometry detection of cell-surface 4D9 (red) binding and isotype control (blue) on primary wild-type and *Trem2*$^{-/-}$ mouse microglia. Data are shown as mean fluorescent intensity (MFI).

B   Uptake assay for fluorescently labeled myelin, Aβ(1–42), and inactivated *Escherichia coli* particles. The top row shows that myelin and Aβ (pseudocolored in white) accumulate within the plasma membrane of primary microglia (labeled with DyLight 649 isolectin and pseudocolored in red). Overnight, pre-treatment with antibody 4D9 significantly increased the percentage of substrate-positive cells (middle row). Pre-treatment with the isotope control did not affect the uptake rate (bottom row). No changes in the uptake of *E. coli* particles were observed. Substrate uptake in the absence of antibody is shown in the top row. Hoechst 33342 was used to counter stain the nuclei (in cyan) and to assess cell density. Scale bar = 20 μm.

C–E Quantification of the change in uptake of myelin (C), Aβ(1–42) (D), and *E. coli* (E) upon antibody treatment relative to the uptake of the respective substrate in the absence of antibody. The number of substrate-positive cells relative to the total number of cells was quantified in each condition. Data represent the mean ± SEM ($n$ = 3). Two-way ANOVA (myelin), Sidak's multiple comparisons test (time effect: $F_{1,4}$ = 0.01123, $P$ = 0.9207; treatment effect: $F_{1,4}$ = 27.98, $P$ = 0.0061; time x treatment effect: $F_{1,4}$ = 0.5948, $P$ = 0.4836); $P$ (1 h) = 0.1362; $P$ (2 h) = 0.0185; unpaired $t$-test (two-tailed) with Welch's correction (Aβ(1–42) and *E. coli*); $P$ (Aβ(1–42)) = 0.0151; $P$ (*E. coli*) = 0.5751; n.s., not significant.

Data information: Statistical evaluations are displayed as follows: *$P$ < 0.05.

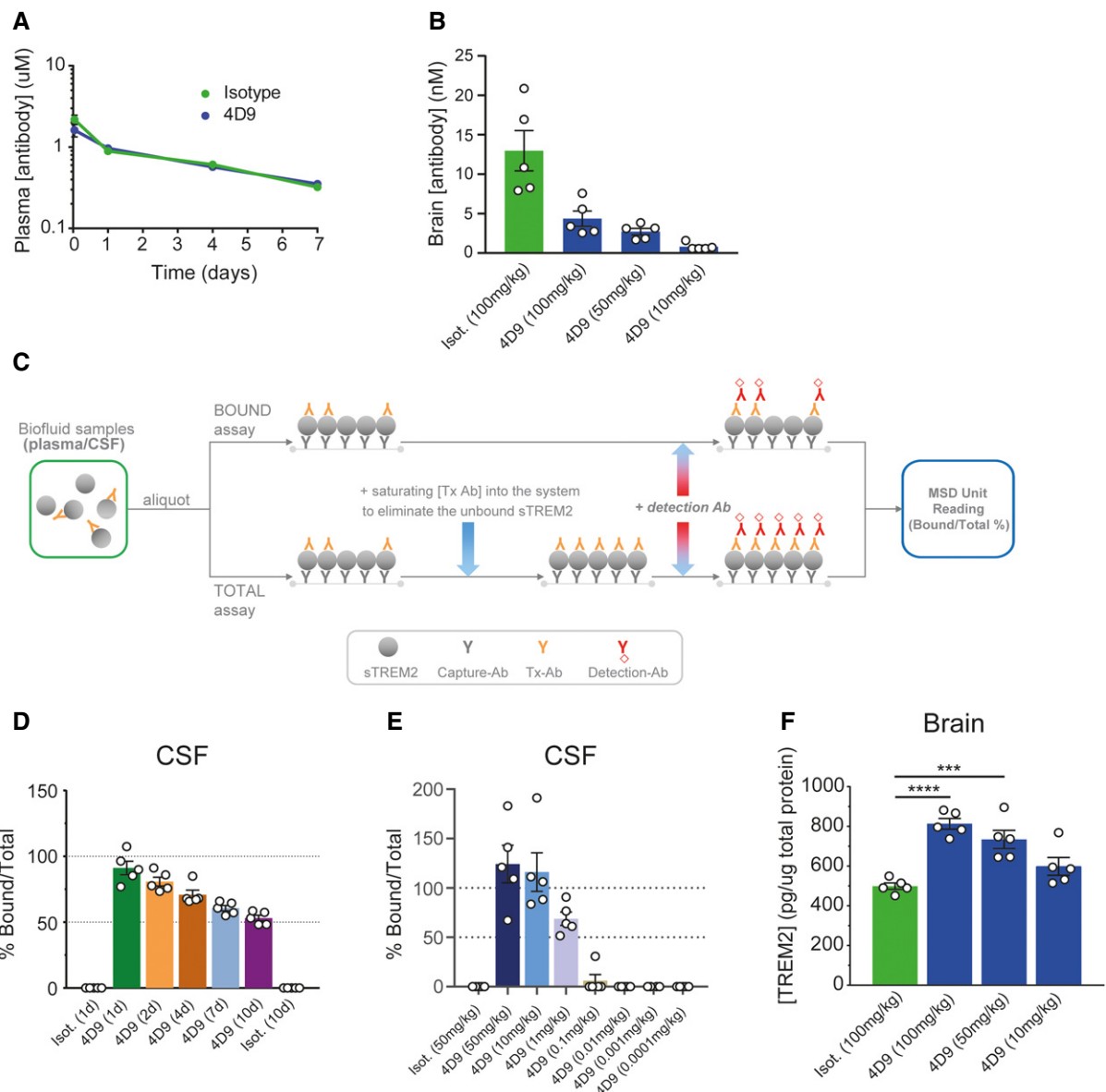

**Figure 5. Pharmacokinetics and *in vivo* target engagement (TE) with 4D9 antibody.**

A  4D9 demonstrates standard IgG pharmacokinetics *in vivo*. Peripheral clearance rates of 4D9 on a human IgG-effectorless backbone compared with an isotype control were determined in wild-type mice by hIgG ELISA detection of plasma antibody concentrations 1 and 24 h, and 4 and 7 days after 10 mg/kg intravenous injection of antibody.

B  Brain antibody concentration was measured 24 h post-intravenous dosing of isotype hIgG at 100 mg/kg, and 4D9-hIgG dosed intravenously at 100, 50, and 10 mg/kg. Detection of hIgG levels by ELISA demonstrated dose-dependent brain concentrations of 4D9 in the single digit nM range. Animals were perfused to minimize IgG contribution from the plasma. Data represent the mean $\pm$ SEM ($n = 5$).

C  Schematic depicting the TE assay setup for bound and total sTREM2 detection. For the bound assay, a secondary anti-human IgG detects 4D9 antibody bound to soluble TREM2 in CSF and plasma. For the total assay, a saturating amount of 4D9 antibody is added to eliminate unbound sTREM2. The ratio of 4D9-bound sTREM2 to total sTREM2 is calculated to determine the level of TE achieved by the concentration of antibody present.

D  Target engagement time course demonstrated near 100% 4D9-bound sTREM2 in CSF of wild-type mice at 24 h post-dose. Over a 10-day time course with time points at days 1, 2, 4, and 7 and study termination at day 10, the bound/total sTREM2 reduces gradually to reach ~ 50% by day 10. Animals were dosed intravenously with 50 mg/kg of isotype and 4D9 antibodies. Data represent the mean $\pm$ SEM ($n = 5$).

E  Target engagement dose response demonstrated saturated bound sTREM2 at 50 and 10 mg/kg with > 50% bound sTREM2 at 1 mg/kg of antibody. 4D9-bound sTREM2 was undetectable at 0.1 mg/kg and lower. 4D9 was IV-dosed at 50, 10, 1, 0.1, 0.01, 0.001, and 0.0001 mg/kg, and isotype-dosed at 50 mg/kg. CSF bound: total sTREM2 in wild-type mice was measured at 24 h. Data represent the mean $\pm$ SEM ($n = 5$).

F  4D9 antibody demonstrates a dose-dependent increase in total brain TREM2 levels. Quantification of total TREM2 in brain lysates from wild-type mice dosed with 4D9 or isotype control was performed by a MSD-platform-based ELISA. Data represent the mean $\pm$ SEM ($n = 5$) and are shown as pg TREM2 per ug of total protein. One-way ANOVA, Dunnett's *post hoc* test, *P* (isotype vs 4D9 [100 mg/kg]) < 0.0001; *P* (isotype vs 4D9 [50 mg/kg]) = 0.0007.

Data information: Only male mice were used. Statistical evaluations are displayed as follows: ***$P$ < 0.001; ****$P$ < 0.0001.

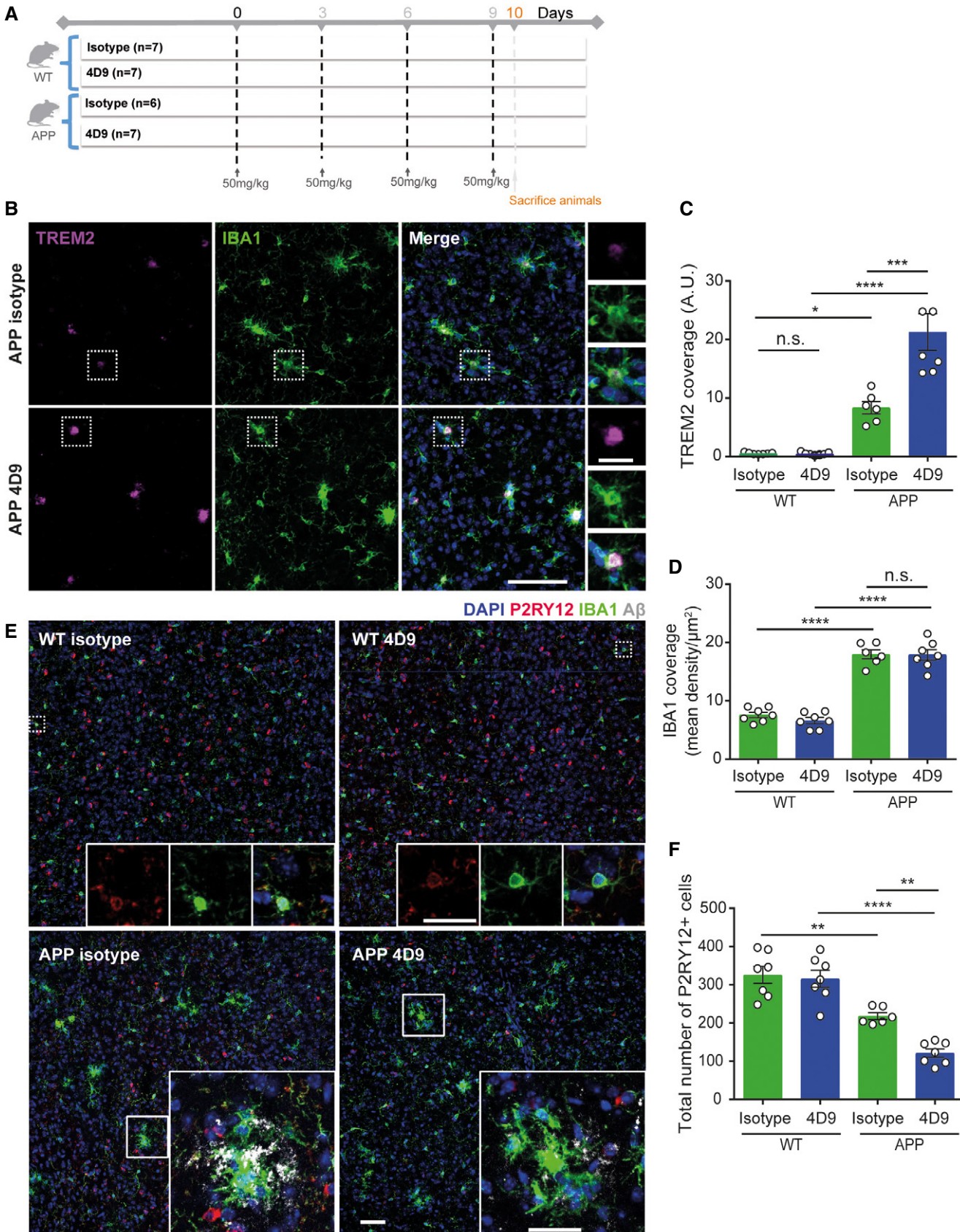

Figure 6.

**Figure 6. 4D9 antibody modulates TREM2 functions within the brain.**

A Schematic outlining study design and timeline of intraperitoneal injections of either isotype control of 4D9 antibody in 6-month-old APP-NL-G-F and age-matched WT mice.

B Immunohistochemical costainings of TREM2 (magenta) and IBA1 (green) microglia in the cortex of isotype control and 4D9-injected WT and APP-NL-G-F mice. Side panel shows images from each staining at a larger magnification indicated by the dotted white boxes. Scale bar = 10 μm; scale bar (inset) = 2.5 μm.

C Quantification of cortical TREM2 stainings shown in (B). Two-way ANOVA, Tukey's multiple comparison test (genotype effect: $F_{1,23}$ = 70.63, $P \leq 0.0001$; treatment effect: $F_{1,23}$ = 14.33, $P = 0.0010$; genotype × treatment effect: $F_{1,23}$ = 14.51, $P = 0.0009$; $P$ (WT isotype vs WT 4D9) > 0.9999; $P$ (WT isotype vs APP isotype) = 0.02; $P$ (WT 4D9 vs APP 4D9) < 0.0001; $P$ (APP isotype vs APP 4D9) = 0.0001; n.s., not significant.

D Quantification of cortical IBA1 staining shown in B. Two-way ANOVA, Tukey's multiple comparison test (genotype effect: $F_{1,21}$ = 215.4, $P \leq 0.0001$; treatment effect: $F_{1,21}$ = 0.5994, $P = 0.4475$; genotype × treatment effect: $F_{1,21}$ = 0.2992, $P = 0.5902$); $P$ (WT isotype vs APP isotype) < 0.0001; $P$ (WT 4D9 vs APP 4D9) < 0.0001; $P$ (APP isotype vs APP 4D9) = 0.9986; n.s., not significant.

E Confocal images of P2RY12 (red), IBA1 (green), and Aβ (gray) costainings from cortex. Top panel: Dotted white boxes indicate the areas in P2RY12 and IBA1 costainings that are magnified as inset. Bottom panel: Insets show P2RY12, IBA1, and Aβ costainings at a larger magnification. Of note, we did not observe a complete co-localization of P2RY12 and IBA1 suggesting different microglial populations. Scale bar = 10 μm; scale bar (insets) = 5 μm.

F Quantification of P2RY12-stained cells in the cortex shown in (E). Two-way ANOVA, Tukey's multiple comparison test (genotype effect: $F_{1,23}$ = 71.99, $P \leq 0.0001$; treatment effect: $F_{1,23}$ = 9.029, $P = 0.0063$; genotype × treatment effect: $F_{1,23}$ = 5.93, $P = 0.0231$); $P$ (WT isotype vs APP isotype) = 0.0018; $P$ (WT 4D9 vs APP 4D9) < 0.0001; $P$ (APP isotype vs APP 4D9) = 0.0051.

Data information: Each data point for all quantifications represents an average of $n = 3$ replicates per mouse. All data represent the mean ± SEM ($n = 6$–7). No gender-specific differences could be observed. Statistical evaluations are displayed as follows: *$P < 0.05$; **$P < 0.01$; ***$P < 0.001$; ****$P < 0.0001$.

2019), and models of retinal degeneration (O'Koren *et al*, 2019). TREM2 plays a central role in the switch of homeostatic to DAM and seems to be acting as a central hub to regulate microglial function (Keren-Shaul *et al*, 2017; Krasemann *et al*, 2017; Ulrich *et al*, 2017). Based on evidence that full-length TREM2 on the cell surface is signaling competent (Kleinberger *et al*, 2014), we hypothesized that strategies that increase the amount of cell-surface TREM2 will increase signaling and enhance TREM2-dependent microglial function. To do so, one could use known ADAM inhibitors or derivates thereof to inhibit shedding of TREM2. However, ADAM10 has numerous physiologically important substrates (Kuhn *et al*, 2016) and additionally promotes Aβ reduction as a non-amyloidogenic protease (Haass & Selkoe, 1993). Thus, inhibition of ADAMs would likely be counterproductive. We therefore investigated whether selectively inhibiting TREM2 shedding with an antibody that hinders ADAM10/17 from accessing the TREM2 cleavage site could increase cell-surface full-length TREM2 and consequently stimulate TREM2-dependent protective phenotypes that for example are reduced by TREM2-associated risk variants. To do so, we screened monoclonal antibodies directed against the entire ectodomain of TREM2 for their ability to bind epitopes within the stalk region, which contains the cleavage site. We identified antibody 4D9, which binds the stalk region in the immediate vicinity of the ADAM10/17 cleavage site and demonstrated increased cell-surface full-length TREM2, and

reduced ADAM10/17-mediated TREM2 shedding. Moreover, the 4D9 antibody concomitantly activated SYK signaling, as detected by phosphorylation of SYK, and enhanced survival of macrophages *in vitro* after growth factor withdrawal, strongly suggesting that indeed full-length, cell-surface TREM2 receptor exerts cell-autonomous signaling functions. Additional evidence for the signaling capacity of full-length TREM2 comes from the analysis of the AD-associated p.TREM2 H157Y variant (Jiang *et al*, 2016) which increases shedding of TREM2 and concomitantly reduces TREM2-dependent phagocytosis (Schlepckow *et al*, 2017), thus furthermore indicating that the amount of cell-surface TREM2 correlates with microglial activity. This is also consistent with our previous finding demonstrating that inhibition of ADAM proteases is sufficient to increase TREM2-dependent phagocytosis (Kleinberger *et al*, 2014).

Similar findings regarding antibody-mediated survival of macrophages were reported by Cheng *et al* (2018), although the mechanism of action and *in vivo* activity of this antibody was not investigated. We now demonstrate that 4D9 exerts its modulating activity via dual mechanisms, first by prevention of shedding and second by agonistic activation of TREM2 receptor signaling. Both mechanisms were found to require bivalent binding in cell-based studies, indicating that cross-linking of cell-surface TREM2 via a stalk epitope underlies antibody mechanism of action. Furthermore, we also provide evidence for *in vivo* TE and modulation of TREM2

**Figure 7. 4D9 antibody reduces plaque burden.**

A, B Overview of the cortex immunohistochemically stained for Aβ plaques in 6-month-old APP-NL-G-F mice treated with isotype control (A) and 4D9 antibody (B). Scale bar = 100 μm.

C Higher magnification images of cortical Aβ plaques in APP isotype and 4D9-injected mice. Scale bar = 10 μm.

D Quantification of cortical plaque area from stainings shown in (C). Mann–Whitney *U*-test, $P = 0.0082$.

E Quantification of total plaque coverage of Aβ stainings in the cortex shown in (C). Mann–Whitney *U*-test, $P = 0.014$.

F Immunoblot analysis of soluble (DEA fraction, top) and insoluble Aβ levels (FA fraction, bottom) in brains of APP-NL-G-F mice treated with either isotype control or 4D9 antibody. While a reduction in levels of soluble Aβ is evident from the DEA fractions, no change in levels of insoluble Aβ could be detected in the FA fractions. Note that the higher background in the DEA immunoblot is due to a much longer exposure time, which was needed to visualize soluble Aβ.

G MSD ELISA quantification of soluble Aβ42 in DEA extracts as shown in (F). Mann–Whitney *U*-test, $P = 0.026$.

H MSD ELISA quantification of insoluble Aβ42 in FA extracts as shown in (F). Unpaired *t*-test (two-tailed) with Welch's correction; $P = 0.0877$.

Data information: Each data point for both quantifications represents an average of $n = 3$ replicates per mouse. All data represent the mean ± SEM ($n = 6$–7). No gender-specific differences could be observed. Statistical evaluations are displayed as follows: *$P < 0.05$; **$P < 0.01$.
Source data are available online for this figure.

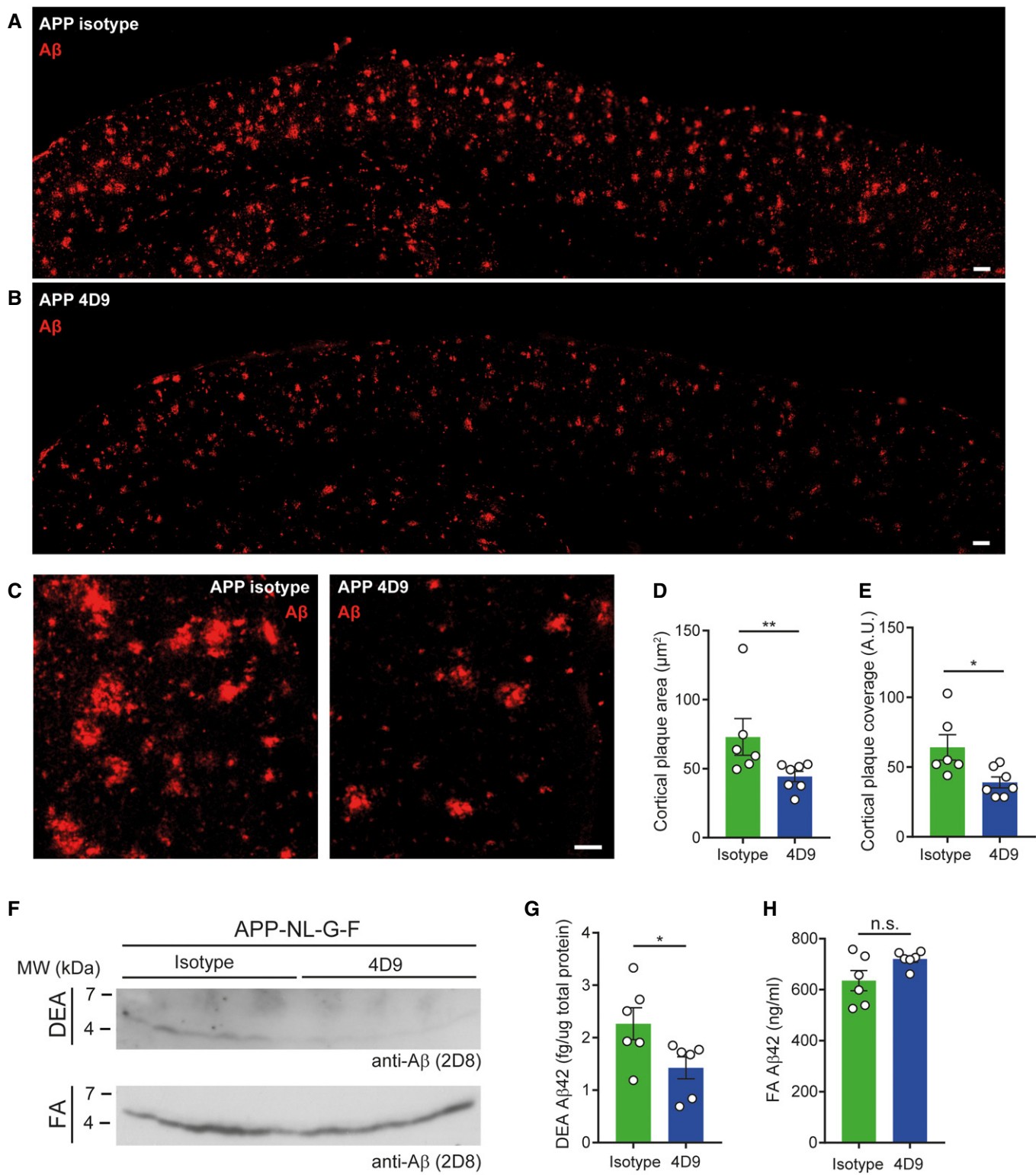

**Figure 7.**

function. *In vivo* TE was successfully demonstrated by near-complete coverage of 4D9-bound sTREM2 in CSF and increased total TREM2 in brain of 4D9-treated mice. Furthermore, increased microglial TREM2 and decreased P2RY12 expression also supports

the immunomodulatory mode of action of 4D9 and suggests that this dual function antibody promotes a shift from homeostasis to a DAM state. This is in line with our observation that antibody 4D9 reduces amyloidogenesis in a mouse model of AD pathology. Since

4D9 appears to selectively reduce soluble Aβ associated with the plaque halo, one may even argue that antibody treatment reduces the amount of toxic Aβ oligomers liberated by amyloid plaques. Reduction in amyloid plaque pathology occurred after a surprisingly short 10-day treatment, which may be related to the diffuse nature of amyloid plaques in this model (preprint: Monasor et al, 2019). However, we currently do not know the fate of engulfed Aβ. Additional data will also be required to demonstrate that antibodies with properties similar to 4D9 can be used to modulate microglial function for the treatment of AD or other indications. In that regard, it is currently unclear whether sTREM2 itself may have essential functions in either a cell-autonomous manner or a non-cell-autonomous manner (Ulland et al, 2017; Zhong et al, 2017, 2019). Moreover, although a loss of function as caused by certain risk variants of TREM2 disables microglia, relegating them to a homeostatic state (Mazaheri et al, 2017), absence of progranulin (PGRN) has the opposite consequence, namely arresting microglia in a disease-associated state (Gotzl et al, 2019). Strikingly, both opposite phenotypes are deleterious to brain health, and absence of either TREM2 or PGRN causes similar brain-wide disturbances in energy metabolism as measured by FDG-PET (Gotzl et al, 2019).

Careful investigation of how Trem2 activity over time correlates with microglial function and additional data elucidating the impact of TREM2 antibody immunomodulatory activity on microglia in vivo will further improve our understanding of the chronic effects of enhanced TREM2 function and potential for efficacy in the clinic. Although TE in brain is possible with 4D9 at early time points, high-dose levels are required to maintain this effect, suggesting TREM2-enhancing antibodies would benefit from blood–brain barrier penetrating technology to improve brain exposure (Yu et al, 2011; Niewoehner et al, 2014). Finally, any TREM2-based therapeutic strategy must take peripheral immune cells into account, since they also express TREM2 and may thus be modulated. In fact, recent work demonstrated TREM2-dependent protective functions in lipid-associated peripheral macrophages, showing effects on lipid uptake and metabolism that are important for the prevention of adipocyte hypertrophy under conditions of a high-fat diet (Jaitin et al, 2019). Strikingly, such protective functions by lipid-associated macrophages may also be boosted by TREM2 function enhancing antibodies.

Although a number of questions remain to be answered, our data suggest that TREM2-modulating approaches can impact disease-relevant microglial functions. Studies from multiple groups have recently improved our understanding of microglial function in disease and how it is impacted by genetic risk factors. Notably, lipid metabolism, storage, and processing has emerged as a potentially important contributor to proper microglial function in the CNS (Chan et al, 2012; Wang et al, 2015; Keren-Shaul et al, 2017; Griciuc et al, 2019; Marschallinger et al, 2020; Nugent et al, 2019). This may indicate that TREM2-stimulating therapeutic approaches could be employed in a variety of different clinical syndromes including AD, frontotemporal lobar degeneration, amyotrophic lateral sclerosis, retinal degeneration, multiple sclerosis, and obesity-associated metabolic syndromes. Furthermore, our earlier findings that antibody-mediated amyloid plaque clearance is TREM2-dependent (Xiang et al, 2016) open the opportunity of a combinatorial treatment using anti-Aβ antibodies together with

microglia stimulating antibodies such as 4D9. This may be specifically of interest in light of the recent findings that a clinical trial using the anti-Aβ antibody aducanumab, though controversial, suggests that treatment may slow clinical decline of patients at the highest dose of the drug. A combinatorial treatment strategy may therefore boost amyloid clearance via enhancement of TREM2-dependent microglial activity and thus enable lower dosing of the anti-Aβ antibody.

# Materials and Methods

### Generation of antibodies specific for mouse TREM2

Eight-month-old female Wistar rats were immunized subcutaneously (s.c.) and intraperitoneally (i.p.) with a mixture of 70 μg recombinant his-tagged mouse Trem2 protein (aa19–171, Creative BioMart) in 500 μl PBS, 5 nmol CpG2006 (TIB MOLBIOL, Berlin, Germany), and 500 μl Incomplete Freund's adjuvant. After 6 weeks, a boost without Freund's adjuvant was given i.p. and s.c. 3 days before fusion. Fusion of the myeloma cell line P3X63-Ag8.653 with the rat immune spleen cells was performed using polyethylene glycol 1500 according to standard procedure (Kohler & Milstein, 1975). After fusion, the cells were plated in 96-well plates using RPMI 1640 with 20% fetal calf serum, penicillin/streptomycin, glutamine, pyruvate, and non-essential amino acids supplemented with HAT Hybri-Max medium supplement (Sigma). Hybridoma supernatants were tested in an enzyme-linked immunoassay using His-tagged mouse TREM2 protein. After blocking with PBS/2% FCS, hybridoma supernatants were added for 30 min. After one wash with PBS, bound antibodies were detected with a cocktail of HRP-conjugated mAbs against the four rat IgG isotypes. HRP was visualized with ready-to-use TMB substrate (1-Step™ Ultra TMB-ELISA, Thermo). The hybridoma cells of TREM2-reactive supernatants were cloned at least twice by limiting dilution. The IgG subclass was determined in an ELISA with mouse anti-rat kappa light chain antibodies as capture and HRP-coupled mouse anti-rat IgG subclass-specific antibodies for detection.

### Hybridoma antibody sequencing

Total RNA was extracted from the monoclonal antibody-expressing hybridoma cells using Qiagen RNeasy Plus Mini Kit (Qiagen, Germantown, MD) following manufacturer's instructions. The extracted total RNA was used as template for 5′-RACE-ready cDNA synthesis (Takara Bio, Mountain View, CA). The heavy chain variable domain (VH) and the light chain variable domain (VK) of 4D9 were amplified separately by PCR using the 5′-RACE adaptor-specific forward primer, rat IgG, or kappa constant-specific reverse primers (Bradbury, 2010). For VK domain amplification, a peptide nucleic acid oligo (CCTGTGGAGGAGGAG-GATGCT-KK) was used to selectively amplify the functional kappa chain (Cochet et al, 1999). The PCR products of VH and VK were purified and cloned into the pCR-TOPO vector (Invitrogen, Carlsbad, CA). The cloned vector was transformed into TOP10 E. coli (Invitrogen, Carlsbad, CA) by chemical transformation and selected on a LB agar plate containing 100 μg/ml carbenicillin. Sanger sequencing was done on bacteria colonies using the M13

forward and M13 reverse primers to sequence VH and VK of the 4D9 antibody.

## Generation of recombinant anti-TREM2 antibodies

### Effectorless IgG

The polypeptide sequences of VH and VK were codon-optimized for CHO cell expression and synthesized by Integrated DNA Technologies (Coralville, IA). The synthesized VH and VK fragments were cloned into the human IgG1 containing effector knockout mutations (L234A, L235A, and P329G) and human kappa chain expression vectors, respectively. ExpiCHO cells (Gibco, Carlsbad, CA) were transfected with the antibody expression plasmids. The culture medium containing the expressed recombinant antibody was harvested 5 days after transfection. The recombinant antibody was purified by MabSelect SuRe (GE Life Sciences, Pittsburgh, PA). The purity of 4D9 recombinant antibody was analyzed by SDS–PAGE (NuPAGE 4–12% Bis–Tris, Invitrogen, Carlsbad, CA) and size exclusion chromatography (Tosoh TSKgel, Tosoh Biosciences, Japan).

### Fab fragment

To generate a monovalent 4D9 Fab antibody fragment, the heavy chain expression vector was modified to include only the VH-CH1 domains as a Fab-heavy chain expression vector. ExpiCHO cells (Gibco, Carlsbad, CA) were transfected with the antibody expression plasmids. The culture medium containing the expressed recombinant antibody or Fab antibody fragment was harvested 5 days after transfection. The recombinant mouse–human chimeric IgG antibody was purified by MabSelect SuRe (GE Life Sciences, Pittsburgh, PA). The recombinant Fab antibody fragment was purified by KappaSelect (GE Life Sciences). The purity of 4D9 recombinant antibody and Fab antibody fragments was analyzed by SDS–PAGE (NuPAGE 4–12% Bis–Tris, Invitrogen, Carlsbad, CA) and size exclusion chromatography (Tosoh TSKgel, Tosoh Biosciences, Japan).

## Epitope mapping of antibody 4D9

Biotinylated polypeptides of the full-length mouse Trem2 stalk (extracellular domain [ECD] amino acids 131–169: CDPLDDQDAGD LWVPEESSSFEGAQVEHSTSRNQETSFPP), tiled stalk regions of the mouse TREM2 (DPLDDQDAG, DDQDAGDLW, DAGDLWVPE, DLWVPEESS, VPEESSSFE, ESSSFEGAQ, SFEGAQVEH, GAQVEHSTS, VEHSTSRNQ, STSRNQETS, RNQETSFPP), and a truncated cleavage site peptide (149–163, CSFEGAQVEHSTSRNQ) were purchased from Elim Biopharmaceuticals, Inc. (Hayward, California, USA). N-terminal cysteine was added to both full-length and truncated stalk peptides to enable maleimide–thiol conjugation of biotin. The lyophilized biotinylated peptides were reconstituted in 20 mM Tris buffer at pH 8.0. Antibody binding to TREM2 stalk peptides was detected using a sandwich ELISA. Briefly, a 96-well half-area ELISA plate was coated with streptavidin overnight at 4°C. The following day, biotinylated TREM2 stalk peptides diluted in 1% BSA/PBS were added to the plate and incubated for 1 h. Antibodies diluted in 1% BSA/PBS were then added and incubated for 1 h. Antibodies bound to peptide were detected with anti-Human Kappa-HRP secondary antibody diluted in 1% BSA/PBS. Plates were developed

with the addition of TMB substrate and stopped by the addition of 2N sulfuric acid.

## Competition peptide ELISA

Antibody binding to the TREM2 stalk peptide (DPLDDQDAGDLWV-PEESSSFEGAQVEHSTSRNQETSFPP-Biotin) was detected using a competitive binding ELISA format assay. A 96-well half-area ELISA plate (Corning Costar #3690) was coated with murine TREM2 His-tagged protein overnight at 4°C. The plate was blocked with a 3% BSA/TBST solution and incubated for 2 h at room temperature. Anti-TREM2 antibody was pre-incubated with a serial dilution of murine TREM2 stalk peptide for 1 h. After blocking, the pre-incubated antibody and peptide sample was added to the plate and incubated for an additional hour. Antibodies bound to murine TREM2 His-tagged protein were detected with anti-human kappa-HRP secondary antibody diluted in 3% BSA/TBST. Plates were developed with the addition of TMB substrate and stopped by the addition of 2N sulfuric acid.

## cDNA constructs

gBlock gene fragments (Integrated DNA Technologies) coding for DAP12-P2A-TREM2 were cloned into the pcDNA5/FRT/TO vector. The coding sequence of murine wild-type DAP12 included a C-terminal V5-epitope tag (5′-GGTAAGCCTATCCCTAACCCTCTC CTCGGTCTCGATTCTACG-3′; aa sequence: GKPIPNPLLGLDST), and the murine wild-type TREM2 sequence included an N-terminal HA-epitope tag (5′-TACCCATACGATGTTCCAGATTACGCT-3′; aa sequence: YPYDVPDYA). The HA-epitope tag was inserted directly after the sequence of the endogenous TREM2 signal peptide (aa1–18). The coding sequences of DAP12 and TREM2 are separated by the P2A sequence (GGAAGCGGAGCTACTAACTTTAGCTTGTTGAA ACAGGCTGGGGATGTAGAGGAAAACCCTGGCCCA) (Kim et al, 2011).

## In vitro TREM2 peptide cleavage detection by fluorescence polarization

TREM2 full length "long" stalk and truncated cleavage site "short" stalk peptide were fluorophore-labeled via addition of a lysine residue to the C-terminal end of the TREM2 stalk peptides (sequences under "Epitope mapping of antibody 4D9") in order to enable conjugation of 5-Carboxyfluorescein. The peptide synthesis, conjugation, and purification was performed by Elim Biopharmaceuticals, Inc. (Hayward, CA, United States). FAM-conjugated short- and long-stalk region mouse TREM2 peptides were prepared in assay buffer (25 mM Tris pH 7.5, 2.5 μM $ZnCl_2$, 0.005% Brij-35) with streptavidin. TREM2 antibodies were then pre-incubated with TREM2 stalk peptides for 30 min at RT. After pre-incubation, ADAM17 was added and incubated with peptides for 20 h at 37°C. The following day, samples were further diluted in assay buffer and transferred to a black opaque 384-well plate. Fluorescence polarization of peptides was subsequently measured on Perkin Elmer EnVision plate reader.

## HEK293 cell culture and generation of isogenic cell lines

Flp-In™ 293 cells (HEK293 Flp-In; Life Technologies) were cultured in Dulbecco's modified Eagle's medium (DMEM) with GlutaMAX I,

supplemented with 10% (v/v) fetal calf serum (FCS), and 200 μg/ml Zeocin and penicillin/streptomycin (PAA Laboratories, Pasching, Austria). Transfections of cDNA constructs were carried out using Lipofectamine 2000 reagent according to the manufacturer's recommendations. For stable overexpression of Dap12-P2A-Trem2 cDNA constructs, HEK293 Flp-In cells were co-transfected with pOG44 (Flp-recombinase expression vector; Life Technologies) and selected using 200 μg/ml hygromycin B. Pools of isogenic cells were used for all experiments.

### Cell culture of bone marrow-derived macrophages

BMDMs were cultured as described before (Xiang *et al*, 2016; Kleinberger *et al*, 2017). For immunoblot analyses, BMDMs were harvested after 7 days, while for all other experiments BMDMs were harvested after 5 days.

### Isolation and culturing of primary microglia for FACS analysis of 4D9 binding

Primary mouse microglia were obtained from mixed glial culture. Briefly, cerebral cortices were isolated from postnatal day 2 C57BL/6 mice obtained from Charles River Laboratories International, Inc. (Davis, CA). Tissues were minced and incubated in 0.25% trypsin-EDTA, mechanically triturated in DMEM/F-12 with 10% FBS, and plated on poly-L-lysine-coated 75-ml flasks. Cultures were maintained for 14 days with media changes every 4 days. Microglial cells were isolated by shaking flasks at 200 rpm in a Lab-Line™ Incubator Shaker for 4 h.

### Antibody treatments, sample preparation, and immunoblotting

HEK293 Flp-In cells stably overexpressing murine Dap12-P2A-Trem2 cDNA constructs were seeded on poly-L-lysine (10 μg/ml)-coated six-well plates at a density of $1.5 \times 10^5$/cm$^2$ and grown for 24 h. Medium was changed to complete culture medium (DMEM, 10% FCS, Pen/Strep) containing 20 μg/ml TREM2 antibodies and incubated for 18–20 h. As control for blocking of Trem2 shedding, cells were treated with GM6001, a broadband ADAM inhibitor (25 μM, Enzo Life Sciences) or DMSO as a vehicle control. Conditioned medium was collected, immediately cooled down on ice, centrifuged at 16,840 *g* for 20 min at 4°C to remove cellular debris, and supernatants were frozen at −20°C until analysis. To prepare membrane fractions, cells were washed twice with ice-cold PBS, resuspended in ice-cold hypotonic buffer (0.01 M Tris, pH 7; 1 mM EDTA; 1 mM EGTA) freshly supplemented with protease inhibitor (Sigma), and incubated on ice for 30 min. After snap freezing in liquid nitrogen and thawing, the disrupted cells were centrifuged at 16,090 *g* for 45 min at 4°C. The resulting pellet was resuspended in STE lysis buffer (150 mM NaCl, 50 mM Tris–HCl, pH 7.6, 2 mM EDTA, 1% Triton X-100), incubated for 20 min on ice, and clarified by centrifugation at 16,090 *g* for 30 min at 4°C. Protein concentrations were measured using the BCA method, and equal amounts of protein were mixed with Laemmli sample buffer supplemented with beta mercaptoethanol. Proteins were then separated by SDS–PAGE and transferred onto polyvinylidene difluoride membranes (Hybond P; Amersham Biosciences, Aylesbury, UK). Bound antibodies were visualized by corresponding HRP-conjugated secondary antibodies

using enhanced chemiluminescence technique (Pierce). BMDMs were seeded at a density of $1.3 \times 10^5$/cm$^2$ in six-well plates, and the medium was supplemented with either 4D9 or isotype control antibody at 20 μg/ml. Conditioned media and cells were collected and processed as described above for HEK293 Flp-In cells after overnight incubation.

Specificity of antibody 4D9 for mouse TREM2 was demonstrated by immunoblotting of 10 ng mouse TREM2-(His)$_6$ as purchased from Sino Biological. For immunoblotting of 10 ng human TREM2-(His)$_6$ (Sino Biological) and 10 ng mouse TREM1-Fc (R&D Systems), AF1828 and AF1187 antibodies (both R&D Systems) were used, respectively. A complete list of all antibodies used is shown in Table EV1.

### ELISA-based quantification of TREM2 protein species

#### *In vitro* cell culture

To quantify the levels of sTREM2 and membrane-bound TREM2 upon treatment with anti-TREM2 antibodies, an electrochemiluminescence-based assay using the Meso Scale Discovery Platform was established. Streptavidin-coated 96-well plates were blocked overnight at 4°C in blocking buffer (3% bovine serum albumin [BSA] and 0.05% Tween 20 in PBS, pH 7.4). Plates were incubated for 1 h at RT with 0.125 μg/ml biotinylated polyclonal sheep anti-TREM2 capture antibody (BAF1729; R&D Systems) diluted in blocking buffer followed by washing two times with washing buffer (0.05% Tween 20 in PBS, pH 7.4) using a plate washer. Samples were mixed 9:1 with denaturation buffer (200 mM Tris–HCl pH 6.8, 4% (w/v) SDS, 40% (v/v) glycerol, 2% (v/v) β-mercaptoethanol, 50 mM EDTA) and boiled at 95°C for 5 min to dissociate TREM2 antibodies from either sTREM2 or membrane-bound TREM2. In the following, plates were incubated at RT for 2 h with the respective samples diluted in sample buffer (1% BSA and 0.05% Tween 20 in PBS, pH 7.4). For quantification of sTREM2, samples were diluted 1:20 and 1:2 for HEK293 Flp-In cells and BMDM, respectively. For quantification of membrane-bound TREM2, 1.5 μg and 1.9 μg from membrane fractions of HEK293 Flp-In cells and BMDM, respectively, were employed. Plates were washed two times with washing buffer before incubation for 1 h at RT with 1 μg/ml rat monoclonal anti-TREM2 antibody (clone 5F4 (Xiang *et al*, 2016)) diluted in blocking buffer. After two additional washing steps, plates were incubated with a SULFO-TAG-labeled anti-rat secondary antibody (1:1,000; Meso Scale Discovery) diluted in blocking buffer and incubated for 1 h at RT. Lastly, plates were washed two times with washing buffer and two times with PBS followed by adding 1× Meso Scale Discovery Read buffer. The light emission at 620 nm after electrochemical stimulation was measured using the Meso Scale Discovery Sector Imager 2400 reader (MSD). Concentrations of sTREM2 and membrane-bound TREM2 were finally calculated using the MSD Discovery Workbench software (v4.0.12). As a standard, we used recombinant murine TREM2 (Hölzel Diagnostika). For standard curve determination, twofold serial dilutions were performed in either sample dilution buffer (sTREM2) or the lysis buffer as used for preparation of membrane fractions (membrane-bound TREM2) supplemented with protease inhibitor mix (Sigma) on the plate spanning concentrations from 4,000 to 62.5 pg/ml. Standard samples underwent the same denaturation procedure as the samples collected from HEK293 Flp-In cells and BMDM. In all incubation

steps at RT, the plate was usually shaken at 300 rpm. All plate washings were performed using a plate washer (ELx405, BioTek).

### In vivo

Mouse brains were perfused with cold PBS, and cortex and hippocampus were sub-dissected out of whole brain tissue and weighed to match tissue mass for each animal. Brain lysate was generated using lysis buffer (Cell Signaling #9803) containing protease inhibitor cocktail (Roche #4693159001) and PhosSTOP (Roche #4906837001) each added into the sample tube (1 ml lysis buffer for 100 mg tissue). Subsequently, one 3-mm stainless steel bead (Qiagen) was added into each tube and samples were homogenized with the Qiagen Tissue-Lyser II at 28 Hz for 3 min. The homogenization step was repeated 3 times. The homogenized samples were then incubated on ice for 10 min followed by centrifugation at 18,660 $g$ for 20 min at 4°. Supernatants were transferred to new tubes and stored at −80° for following protein concentration determination.

To quantify total brain TREM2 levels, an electrochemiluminescence-based assay using the Meso Scale Discovery Platform was utilized. Streptavidin-coated 96-well plates were blocked using blocking buffer (25% MSD Blocker A and 75% TBST) for 1 h at R.T. Plates were then incubated for 1 h at RT with 1 µg/ml biotinylated polyclonal sheep anti-TREM2 capture antibody (BAF1729; R&D Systems) diluted in blocking buffer followed by washing three times with washing buffer (0.05% Tween 20 in TBS) using a plate washer. Brain lysates were diluted in blocking buffer (1:5) and aliquoted on MSD-compatible plates. All plates were incubated at RT for 1 h. Plates were washed three times with washing buffer on a ELx406 plate washer (BioTek) before incubation for 1 h at RT with 10 µg/ml monoclonal anti-TREM2 antibody (clone 4D9) diluted in blocking buffer. Plates were shaken at 800 rpm in all incubation steps.

After three additional washing steps, the MSD plates were incubated with SULFO-TAG-labeled anti-human secondary antibody (1 µg/ml; Meso Scale Discovery) diluted in blocking buffer and incubated for 1 h at RT. Lastly, plates were washed three times with washing buffer followed by adding 2× Meso Scale Discovery Read buffer. The light emission at 620 nm after electrochemical stimulation was measured using the Meso Scale Discovery Sector Imager S600 reader (MSD). Concentrations of TREM2 were calculated using the MSD Discovery Workbench software. Recombinant murine extracellular domain TREM2 protein was used to generate a standard curve. For standard curve determination, fourfold serial dilutions were performed in lysis buffer as used for preparation of brain lysates supplemented with protease inhibitor mix (Sigma) on the plate with concentrations ranging from 62.5 ng/ml to 15.25 pg/ml. Data are normalized per ug total protein.

### Dose-dependent sTREM2 shedding assay

HEK-293F cells stably overexpressing mouse TREM2 were plated at 50,000 cells per well in complete media (DMEM supplemented with GlutaMAX, 10% FBS, 1% Pen Strep) in PDL-coated plates and incubated for 24 h at 37°C. The following day, media were removed and cells were washed once with Opti-MEM and treated with a dose titration of 4D9 antibody diluted in Opti-MEM and incubated for 18 h at 37°C. After incubation, supernatants were harvested for detection of shed mouse sTREM2 in cell culture supernatant.

To quantify the levels of sTREM2 upon treatment with anti-TREM2 antibodies, an electrochemiluminescence-based assay using the Meso Scale Discovery Platform was established. Streptavidin-coated 96-well plates were incubated for 2 h at RT with 1 µg/ml biotinylated polyclonal sheep anti-TREM2 capture antibody (BAF1729; R&D Systems) diluted in PBS (pH 7.4). After washing two times with washing buffer (0.05% Tween 20 in TBS, pH 7.4) using a plate washer, plates were blocked for 1 h at RT in blocking buffer (3% BSA and 0.05% Tween 20 in TBS, pH 7.4). Samples were mixed 13:7 with denaturation buffer (1× NuPAGE Sample Reducing Agent, 1× NuPAGE LDS Sample Buffer) and boiled at 95°C for 5 min to dissociate TREM2 antibodies from either sTREM2 or membrane-bound TREM2. In the following, plates were incubated at RT for 2 h with the respective samples diluted 1:10 in blocking buffer (3% BSA and 0.05% Tween 20 in TBS, pH 7.4). Plates were washed two times with washing buffer before incubation for 1 h at RT with sulfo-tagged polyclonal sheep anti-TREM2 (AF1729; R&D Systems) diluted 1:5,000 in blocking buffer. Lastly, plates were washed two times with washing buffer and followed by the addition of 2× Meso Scale Discovery Read buffer. The light emission at 620 nm after electrochemical stimulation was measured using the Meso Scale Discovery Sector Imager 2400 reader (MSD). Concentrations of sTREM2 were finally calculated using the MSD Discovery Workbench software (v4.0.12). As a standard, we used recombinant murine TREM2 (Hölzel Diagnostika). For standard curve determination, half-log-fold serial dilutions were performed in sample dilution buffer (sTREM2) on the plate spanning concentrations from 4,000 to 62.5 pg/ml. Standard samples did not undergo denaturation procedure prior to incubation on the plate. In all incubation steps at RT, the plate was usually shaken at 600 rpm. All plate washings were performed using a plate washer (ELx405, BioTek).

### FACS for cell-surface 4D9 binding to mouse TREM2

For mouse TREM2 surface staining, BMDM and mouse microglial cells were harvested, pelleted, resuspended in FACS buffer (PBS + 0.5% BSA + FcX solution 5 µl/ml) at $10^6$/ml, and seeded at 100,000 cells per well in a 96-well round bottom. Cells were centrifuged, and biotinylated-4D9 and isotype control were added at 200 nM and incubated for 45 min on ice. Cells were centrifuged and washed with FACS buffer for three times. Cells were then incubated with secondary antibody (APC Streptavidin 1:300, BD Pharmingen #554067) for 30 min on ice. After incubation, cells were washed with FACS buffer three times and resuspended in 100 µl of FACS buffer, and then analyzed by flow cytometry (BD FACSCanto II).

### Antibody binding kinetics

Human TREM2, mouse TREM2, and mouse TREM1 ECD binding affinities to 4D9 were determined by SPR using a Biacore 8K instrument. Biacore Series S CM5 sensor chip was immobilized with a mixture of two monoclonal mouse anti-Fab antibodies (Human Fab Capture Kit from GE Healthcare). Serial twofold dilutions of recombinant mouse Trem2 were injected at a flow rate of 30 µl/min for 300 s followed by 1,200-s dissociation in HBS-EP+ running buffer (GE, #BR100669). Binding response was corrected by subtracting the RU from a blank flow cell. A 1:1 Langmuir model of simultaneous fitting of $k_{on}$ and $k_{off}$ was used for kinetics analysis.

## Immunocytochemistry of HEK293 cells

HEK Flp-In cells stably overexpressing either empty vector or mouse TREM2/DAP12 were seeded on coverslips coated with poly-L-lysine (10 µg/ml). After incubation at 37°C and 5% $CO_2$ for 24–48 h, media were removed and cells were washed three times for 5 min with PBS. Cells were incubated with freshly prepared fixation solution (4% (w/v) paraformaldehyde, 4% sucrose in PBS pH 7.4) for 10 min and washed three times for 5 min with PBS. Upon quenching the autofluorescence for 15 min at RT using 50 mM $NH_4Cl$ in PBS, cells were again washed three times for 5 min with PBS. In the following, cells were blocked with 5% BSA in PBS overnight at 4°C. Cells were then incubated for 2 h at 37°C with either antibody 4D9 (0.4 µg/ml) or an anti-HA antibody (clone 3F10; Roche; 1 µg/ml) diluted in 0.5% BSA in PBS. After incubation with the primary antibody, cells were washed three times for 5 min with PBS. Goat anti-rat Alexa 488 secondary antibody (Thermo Fisher Scientific; #11006; 1:500) was added, and cells were incubated for 1 h at 37°C followed by three washes with PBS for 5 min. Cells were then counter-stained with DAPI (1:5,000 in PBS) and washed another three times for 5 min at 37°C. Finally, coverslips were mounted using Fluoromount Aqueous Mounting Medium (F4680, Sigma). Confocal images were acquired using a LSM 800 confocal microscope (Zeiss) and the ZEN 2.5 Zeiss software package. Two–three images were taken per condition using a 63× oil differential interference contrast objective (1.40 Oil DIC M27) at 1,437 × 1,437 pixel resolution with a z-step size of usually 0.7 µm at 4 µm thickness. Image analysis was subsequently done using the FIJI software package (ImageJ). Briefly, channels and focal planes were split followed by background subtraction using a sliding paraboloid (rolling ball radius of 50 pixel) on the focal plane of interest. Brightness and contrast were then adjusted such as to optimize the balance between TREM2 surface and DAPI stainings. Finally, channels were merged again to generate images as shown in Fig 1D. Preparation of coverslips was conducted using sterile filtered buffers.

## p-SYK AlphaLISA

HEK293 Flp-In cells overexpressing mouse TREM2 and mouse DAP12 were plated in complete media (1× DMEM, 1× GlutaMAX, 10% FBS, 1% Pen Strep) at 50,000 cells/well in PLL-coated 96-well plates. Cells were incubated overnight at 37°C in a cell culture incubator. The following morning, plates were washed 3× with HBSS and 50 µl of liposome/antibody mixture or liposome alone diluted in HBSS was added to each well. Antibodies and liposomes were employed at 20 µg/ml and 1 mg/ml, respectively. Cells were incubated for 5 min at 37°C before treatment solutions were removed from the plate by gentle pipetting. Cells were quickly lysed with 40 µl lysis buffer supplemented with protease inhibitor mix (Sigma) and phosphatase inhibitor (PhosSTOP, Roche) and incubated on ice for 30 min. Phosphorylated SYK (p-SYK) was measured in cell lysates using an AlphaLISA SureFire Ultra p-SYK Assay Kit (Tyr525/526; PerkinElmer), and plates were read on an EnSpire Multimode Plate Reader (PerkinElmer). To evaluate effects of antibody 4D9 alone on p-SYK levels, the experiment was carried out identically with the exception that washing steps employing HBSS were omitted.

## Preparation of liposomes

POPC/POPS (7:3) liposomes at 10 mg/ml were prepared as follows: 7 mg POPC and 3 mg POPS were dissolved in chloroform followed by thorough evaporation of solvent. The lipid mixture was then resuspended in 1 ml HBSS, and suspensions were extruded using 100 nm polycarbonate membranes (Whatman, article no. 800309) and a LiposoFast extruder device (Sigma-Aldrich) to generate large unilamellar vesicles.

## Cell survival

Bone marrow-derived macrophages were washed and resuspended in culture media and plated in a 96-well plate that was pre-coated with a titration of mouse Trem2 antibody clone 4D9 (hIgG1-effector-less) or isotype control overnight at 4°C. Antibody-coated plates were washed with PBS twice, and cells were plated in media with a low concentration of M-CSF (5 ng/ml, Gibco, Cat# PHC9501). After 5 days, cellular ATP levels were measured by luminescence detection to indicate cell viability with CellTiter-Glo (Promega Catalog Number G7571).

## Myelin, Aβ, and *E. coli* phagocytosis by primary microglia

Postnatal microglia were isolated with the Neural Tissue Dissociation Kit (Miltenyi Biotec, Cat# 130-092-628) and anti-CD11b magnetic beads (Miltenyi Biotec, Cat# 130-049-601) according to the manufacturer's instructions. Briefly, brains from p7-p10 animals were dissociated to a single-cell suspension, which was then passed through a 70-µm strainer. The cells were then washed by repeated centrifugation at 300 g and incubated with the CD11b$^+$ beads for 15 min on ice. The suspension was then passed through the LS Column (Miltenyi Biotec, Cat# 130-042-401) on the magnet to isolate CD11b$^+$ cells. The cells were plated in growth media: DMEM supplemented with high glucose, sodium pyruvate, penicillin/streptomycin, 10% fetal calf serum, and 10% of media conditioned by the CSF1-expressing l,929 cells. The cells were allowed to grow to full confluence. On the day before the experiment, the media were changed to TIC media: DMEM containing human TGF-β2 (2 ng/ml, PeproTech, Cat# 100-35B), murine IL-34 (100 ng/ml, R&D Systems, Cat# 5195-ML-010), and cholesterol (1.5 µg/ml, Avanti Polar Lipids, Cat# 700000p-1g) (Bohlen *et al*, 2017). Treatment with 4D9 and isotype control antibodies (both at 20 µg/ml) was performed overnight in TIC media. The various substrates were added for the indicated times before fixation in 4% paraformaldehyde for 15 min.

Myelin was labeled with the PKH26 kit (Sigma, Cat# PKH26GL-1KT). Briefly, 250 µg of myelin was resuspended in solution C from the kit and the dye PKH26 was added to the mix. After 5 min, myelin was centrifuged and washed in sterile PBS. One micromolar human Aβ(1–42)-HiLyte (Anaspec Inc.) and *E. coli* pHrodo (Thermo Fisher Scientific) were prepared as previously described (Kleinberger *et al*, 2014). The cells were stained with isolectin conjugated to DyLight 649 (Vectorlabs, Cat# DL-1208) for 60 min at room temperature and with Hoechst 33342 for 10 min before mounting. The stained cells were then imaged on a Leica sp5 confocal microscope with a 63× objective. At least three fields per condition were recorded, and the number of substrate-positive cells was normalized to the total number of cells. The cells treated with each specific

substrate in the absence of the antibody were used as the baseline for the same treatment in the presence of the 4D9 or the RSV isotype control antibody. The final graph shows the average increment in the uptake of various substrates after antibody treatment in three independent experiments.

### Mouse lines

Wild-type C57BL/6J mice used for pharmacokinetic analysis of antibodies were purchased from the Jackson Laboratory at 2 months of age. Wild-type C57BL/6J mice as used for immunoblot and MSD ELISA analyses of BMDM in Fig 3B and C were purchased from Charles River and were 2.5–3 months of age when sacrificed. TREM2 KO mice on C57BL/6J background were originally received from Marco Colonna (Turnbull et al, 2006; Mazaheri et al, 2017). An additional TREM2 KO mouse line used for BMDM and primary microglia culture-related experiments in Figs 3A and D, and 4A was purchased from Jackson Laboratory (Stock#027197) and back-crossed to C57BL/6J to generate TREM2 KO heterozygous mice and then intercrossed the heterozygous mice together to generate TREM2 KO and littermate wild-type control. These mice were at least 2.5 months of age when they were sacrificed. To investigate 4D9 modulatory effects on TREM2 and P2RY12 by immunohisto-chemistry, we used the APP-NL-G-F knock-in mice described by Saido and colleagues (Saito et al, 2014). All mouse husbandry and experimental procedures were approved by Denali Institutional Animal Care and Use Committee. Similarly, all animal experiments at DZNE were performed in accordance with animal-handling laws of the state of Bavaria (Germany). Housing conditions included standard pellet food and water provided *ad libitum*, a 12-h light–dark cycle at temperature of 22°C with maximal 5 mice per cage and cage replacement once per week, and regular health monitoring.

### Dosing APP-NL-G-F mice with 4D9-effectorless antibody

Six-month-old APP-NL-G-F (App$^{tm3.1Tcs}$/App$^{tm3.1Tcs}$) and age-matched C57BL/6J wild-type (WT) mice were randomly allocated to either isotype control or 4D9 antibody treatment group. Both male and female mice were used for either genotype. Mice were weighed and then intraperitoneally injected with 50 mg/kg isotype control or 4D9 antibody on days 0, 3, 6, and 9, and the study was terminated on day 10.

### Antibody pharmacokinetics

C57BL/6J mice were dosed with 4D9-hIgG-effectorless antibody or control IgG at 10 mg/kg through the intravenous (IV) tail vein injection. Blood samples were collected at 1 h, 24 h, day 4, and day 7 after dosing. The in-life blood at the first three time points was collected through the submandibular bleeding, and the terminal blood at day 7 was collected through the cardiac puncture. Blood was collected in EDTA tube (Sarstedt Microvette 500 K3E) with slowly inverting for 10 times and then centrifuged at 15,350 *g* for 7 min at 4°C. Plasma (top layer) was transferred to 1.5-ml Eppendorf tube and stored at −80°C until measurement.

Human IgG antibody concentration was measured using a standard colorimetric ELISA. 384w MaxiSorp plates were coated with donkey anti-human IgG capture antibody (Jackson ImmunoResearch) diluted to 1 µg/ml in sodium bicarbonate and incubated overnight at 4°C. The following morning, plates were washed 3× with PBST and blocked with 5% BSA/PBST solution for 1 h at RT. Plates were washed again before addition of sample. Samples were incubated on the plate for 2 h at RT. To detect, plates were washed and goat anti-human IgG-HRP (Jackson ImmunoResearch) diluted 1:40,000 in 1% BSA/PBST solution was added to each well and incubated for 2 h at RT. Plates were then washed, developed with TMB substrate, and stopped with the addition of 2N sulfuric acid. Absorbance was measured at 450 nm on the BioTek plate reader.

### Target engagement

The levels of bound (B) sTREM2 and total (T) sTREM2 were determined by measuring [4D9-bound ms-sTREM2] and [ms-sTREM2] in samples using a sandwich-layout MSD as previously described. Each biospecimen (plasma, CSF, or brain lysate) utilized the same MSD TREM2 detection formats for the bound and total assay. For the bound assay, [4D9-bound ms-sTREM2] is directly determined by the sulfo-tagged detection-Ab against 4D9 (Tx-Ab). In the total assay, adding additional Tx-Ab to the biofluid samples is required to saturate any remaining unbound fraction of ms-sTREM2 in the biofluids. After this step of Tx-Ab saturation, the total [ms-sTREM2] can be measured by the same Sulfo-tagged Detection-Ab. The target engagement (TE) levels for each biofluid sample can be represented by the percentage of [BOUND sTREM2]/[TOTAL sTREM2] (B/T %).

### Immunofluorescence analyses and confocal imaging of mouse brain sections

Following cardiac perfusion with ice-cold PBS, left-brain hemispheres were immerse-fixed in 4% paraformaldehyde for 24 h and in 30% sucrose for 24 h subsequently. After freezing, 50-µm micro-tome-cut free-floating sections were washed briefly and then blocked using 5% donkey serum for 1 h at room temperature. Subsequently, sections were incubated with primary antibodies (IBA1 1:300, Invitrogen, PA5-27436; Aβ$_{1-40}$ 1:3,000, clone 3552 (Page et al, 2010), in-house antibody) at 4°C overnight or (TREM2 1:50, R&D Systems, AF1729; P2RY12 1:100, BioLegend, clone S16007D) over two nights at 4°C. Sections were washed and incubated in donkey anti-rabbit AF488 (1:1,000, Invitrogen, A32790) and AF647 (1:1,000, Invitrogen, A32795), donkey anti-rat AF488 (1:500, Invitrogen, A21208), and donkey anti-sheep AF555 (1:500, Invitrogen, A21436) secondary antibodies for 2 h at room temperature. Lastly, slides were washed and stained with 4′,6-diamidino-2-phenylindole (DAPI, 5 µg/ml) before mounting coverslips with ProLong™ Gold antifade reagent (Thermo Fisher Scientific). For all stainings, an average of three sections per mouse was stained and quantified. Confocal images were acquired using a LSM 800 confocal microscope (Zeiss) and the ZEN 2.5 Zeiss software package. At least three images were taken to include larger areas of the cortex using 10× (Plan-Apochromat 10×/0.25) and 20× (Plan-Apochromat 20×/0.8) objectives at 2,048 × 2,048 pixel resolution with a z-step size of 1.1 µm at 18 µm thickness. Leica DMi8 fluorescence microscope and Leica Application Suite X software were used to acquire tile scan images of the entire cortex using 10× objective (Plan-Apochromat 10×/0.32 PH1). All P2RY12-positive cells per image were counted on FIJI software

(ImageJ) using the "Cell counter" plugin. For IBA1, TREM2, and Aβ plaque coverage analyses, confocal acquired images for cortex were imported into FIJI and channels were separated by "Image/Color/ Split Channels". Following this, background noise was removed using Gaussian filtering and intensity distribution for each image was equalized using rolling ball algorithm. All layers from a single image stack were projected on a single slice by "Stack/Z projection". Lastly, the stainings were segmented using automatic thresholding method in FIJI (IBA1, "Moments"; TREM2, "MaxEntropy"; plaques, "Triangle"). Coverage analysis indicates number of stained positive pixels over total number of pixels in a given area. Plaque area was measured in FIJI on thresholded images.

### Biochemical characterization of brain tissue from APP-NL-G-F mice

DEA (0.2% diethylamine in 50 mM NaCl, pH 10, and protease inhibitor mix [Sigma, P8340]) and RIPA lysates (20 mM Tris–HCl [pH 7.5], 150 mM NaCl, 1 mM EDTA, 1 mM EGTA, 1% NP-40, 1% sodium deoxycholate, 2.5 mM sodium pyrophosphate, and protease inhibitor mix [Sigma, P8340]) were prepared from brain hemispheres (Willem *et al*, 2015). RIPA lysates were centrifuged at 14,000 *g* (60 min at 4°C), and the remaining pellet was homogenized in 70% formic acid (FA fraction). The FA fraction was neutralized with 20× 1 M Tris–HCl buffer at pH 9.5 and used for Aβ analysis. For Aβ detection, proteins were separated on Tris-Tricine gels (10–20%, Thermo Fisher Scientific), transferred to nitrocellulose membranes (0.1 μm, GE Healthcare), which were boiled for 5 min in PBS and subsequently incubated with the blocking solution containing 0.2% I-Block (Thermo Fisher Scientific) and 0.1% Tween 20 (Merck) in PBS for 1 h, followed by overnight incubation with 2 μg/ml 2D8 antibody in the blocking solution. The rat monoclonal 2D8 antibody against Aβ was described before (Shirotani *et al*, 2007). Aβ in DEA and FA fractions was quantified by a sandwich immunoassay using the Meso Scale Aβ Triplex plates and Discovery Sector Imager 2400 as described previously (Page *et al*, 2008). Samples were measured in triplicates.

### Statistical analyses

All statistical analyses were performed using GraphPad Prism 7. Data are shown as mean ± SEM or median ± SEM. Whether data were normally distributed was tested using the D'Agostino and Pearson, Shapiro–Wilk, and Kolmogorov–Smirnov normality tests. In case data were not normally distributed, we applied the Mann–Whitney *U*-test as shown in Fig 7D, E and G. In all other data sets, data were distributed normally. We therefore applied either *t*-test or one-way ANOVA. When comparing multiple groups, two-way ANOVA was applied. ANOVA analyses were always followed by appropriate *post hoc* analyses. Regarding animal studies, sample size was determined based on experience from previous findings and animal availability, i.e., no power calculation was done prior to study design. No specific randomization procedure was employed. However, all animals were randomly assorted into treatment groups after genotyping. Samples were pseudoanonymized. None of the animals were excluded from analysis based on the results from ROUT analysis to identify outliers on GraphPad Prism. Only in the data of Fig 3B, we identified one outlier by ROUT analysis which

### The paper explained

#### Problem
Numerous clinical trials with amyloid β-peptide targeting strategies failed to show cognitive improvement and sometimes even caused adverse effects. Novel therapeutic targets and disease-modulating strategies are therefore desperately required for treating neurodegenerative diseases, a significant global unmet medical need. The microglial-expressed triggering receptor expressed on myeloid cells 2 (TREM2) is known to exert protective functions in the CNS, which are diminished by disease-associated loss-of-function risk variants. We therefore sought to identify monoclonal antibodies capable of enhancing TREM2 function on microglia.

#### Results
Monoclonal antibodies raised against the entire ectodomain of mouse TREM2 were screened to identify those which increase TREM2 expression on the cell surface. Antibody discovery efforts identified clone 4D9, with an epitope in the stalk region of the TREM2 extracellular domain, proximal to the ADAM cleavage site. The 4D9 antibody elicited a strong increase in cell-surface TREM2 and concomitantly reduced proteolytic shedding of soluble TREM2 (sTREM2). 4D9 activated TREM2-dependent phospho-SYK signaling in a dose-dependent manner and enhanced TREM2 function resulting in profound survival rescue with restricted M-CSF, increased phagocytosis of Aβ and myelin *in vitro*. *In vivo* target engagement was demonstrated by nearly saturated binding of sTREM2 in CSF and by increased TREM2 and decreased P2RY12 expression in brains of 4D9-dosed mice. Consistent with the induction of protective functions in microglia, amyloidogenesis in a mouse model for Alzheimer's pathology was reduced following 4D9 treatment.

#### Impact
This is the first characterization of a TREM2 antibody's mechanism of action which elicits TREM2-dependent microglial activity. The function enhancing TREM2 antibody binds to the stalk region of TREM2 near the sheddase site and increases TREM2 signaling and cell-surface receptor via bivalent interactions. These results support the therapeutic hypothesis of enhancing TREM2 activity as a novel approach for efficacious interventions for Alzheimer's disease, and potentially other neurodegenerative diseases as well as metabolic disorders associated with obesity.

was therefore excluded from analysis. In the figures, statistical evaluations are displayed as follows: *$P < 0.05$; **$P < 0.01$; ***$P < 0.001$; ****$P < 0.0001$; n.s., not significant ($P > 0.05$). All data were generated from at least three independent biological experiments or three mice per group unless otherwise stated.

**Expanded View** for this article is available online.

### Acknowledgements
The monoclonal antibody facility is supported by the Deutsche Forschungsgemeinschaft (DFG) within the framework of the Munich Cluster for Systems Neurology (EXC 2145 SyNergy; ID 390857198). C.H. is supported by the Koselleck Project HA1737/16-1 of the DFG, and M.S. by the Dr. Miriam and Sheldon G. Adelson Foundation. This project also received funding from the Innovative Medicines Initiative 2 Joint Undertaking under grant agreement No. 115976. This Joint Undertaking receives support from the European Union's Horizon 2020 research and innovation program and EFPIA. We thank Dr. Takaomi Saido for providing access to his mouse model for AD pathology. We also thank Harald Steiner, Anja Capell, Frits Kamp, Pascal E. Sanchez, and Ryan Watts for

critically reading this manuscript. Joseph Duque and Tina Giese contributed to antibody production and quality control of materials for *in vitro* and *in vivo* studies. Lihong Zhan provided cells and reagents for *in vitro* studies. Hoang Nguyen and Timothy Earr helped with *in vivo* studies.

## Author contributions

CH, KS, JWL, and KMM conceived the study and analyzed the results. CH wrote the manuscript with further input from JWL, KMM, GDP, MS, and KS. KS, GK, NP, BB, JS, MW, HH, and RP performed the tissue culture experiments, Western blot analyses, and ELISAs. XX provided BMDM. Monoclonal antibodies were generated by RF and screened by GK, KS, and NP. AS purified monoclonal antibodies from hybridoma supernatants. JIP sequenced and cloned hybridoma antibodies. DJK performed Biacore analyses. CM led antibody production activities. FH advised on TE assay design and execution. DX performed studies in mice, and HS and JS performed FACS analyses. C-CL developed the TE assay and performed B/T% sTREM2 detection. MS and LC-C performed the uptake assays using primary microglia. GW performed the uptake assays using BMDM, and JS performed survival assays in BMDM. BN prepared liposomes and performed ELISAs. ST contributed conceptually to the *in vivo* studies. GDP contributed to *in vivo* studies design. SP performed immunohistochemistry on brain sections.

## Conflict of interest

C.H. collaborates with Denali Therapeutics, participated on one advisory board meeting of Biogen, and received a speaker honorarium from Novartis and Roche. C.H. is chief advisor of ISAR Bioscience. J.W.L., K.M.M., J.I.P., H.S., J.S., R.P., D.X., C.M., D.J.K., C.C.L., F.H., and G.D.P. are employees of Denali Therapeutics.

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
