## [Review Process File · EMBO Molecular Medicine]

Enhancing protective microglia activities with a dual function TREM2 antibody to the stalk region

Kai Schlepckow, Kathryn M. Monroe, Gernot Kleinberger, Ludovico Cantuti-Castelvetri, Samira Parhizkar, Dan Xia, Michael Willem, Georg Werner, Nadine Pettkus, Bettina Brunner, Alice Sülzen, Brigitte Nuscher, Heike Hampel, Xianyuan Xiang, Regina Feederle, Sabina Tahirovic, Joshua I. Park, Rachel Prorok, Cathal Mahon, Chun-Chi Liang, Ju Shi, Do Jin Kim, Hanna Sabelström, Fen Huang, Gil Di Paolo, Mikael Simons, Joseph W. Lewcock & Christian Haass

Review timeline:

Submission date:	30th Jul 2019
Editorial Decision:	18th Sep 2019
Revision received:	17th Jan 2020
Editorial Decision:	6th Feb 2020
Revision received:	12th Feb 2020
Accepted:	17th Feb 2020

Editor: Céline Carret

Transaction Report:

1st Editorial Decision

18th Sep 2019

Thank you for the submission of your manuscript to EMBO Molecular Medicine and for your patience. We have now heard back from the three referees whom we asked to evaluate your manuscript.

You will see from the set of reviews pasted below that the referees find the study interesting and timely, however they also mentioned the limited *in vivo* data, insufficient mechanistic analysis and the need for better antibody characterisation. Upon cross-commenting, it became evident that concentrating on the antibody would be a better choice than providing additional *in vivo* data, as you also implied during our pre-consultation exercise. As such, and after consulting with my colleagues, we would like to invite a revision along the lines that you have proposed in your letter.

We would therefore welcome the submission of a revised version within three months for further consideration and would like to encourage you to address all the criticisms raised as suggested to improve conclusiveness and clarity. Please note that EMBO Molecular Medicine strongly supports a single round of revision and that, as acceptance or rejection of the manuscript will depend on another round of review, your responses should be as complete as possible.

***** Reviewer's comments *****

Referee #1 (Comments on Novelty/Model System for Author):

see comments to authors

however technical quality gets dinged for the statistical analysis of the lipodomic data with no correction for false discovery

Although there is only one other current publication of a TREM2 agonist antibody (there may be more), at least 10 other groups are doing similar things, and there is nothing really special about this antibody at this point...so again somewhat novel but not completely
 Without definitive data for target engagement in vivo the medical impact is unclear.
 The model system issue is more of general issue for the AD field, how would we determine if the antibody is beneficial? Altering some microglial phenotype does not mean the efficacy would be apparent in AD. M

Referee #1 (Remarks for Author):

This paper describes the identification of a TREM2 binding antibody that appears to inhibit shedding of mouse TREM2 and increase the cell surface levels of TREM2. Convincing data is provided that suggest that the antibody shows high affinity and epitope mapping is performed. Short term PK-PD studies are performed as well as in vitro/ex vivo evidence for functional activity following binding to either transfected cells or primary microglial cells. In vivo studies claim to show alterations in abnormal lipid profiles in 5X FAD mice, but these data are problematic due to lack of correction for multiple testing. Based on the hypothesis that partial loss of function of TREM2 underlies Alzheimer's risk associated with TREM2 AD risk variants, a number of groups are isolating potentially TREM2 agonist antibodies. Indeed, a recent report (Cheng JBC 2018 describes at least one). Clearly, a challenge for the field is what are the readouts for successful target engagement of such antibodies when dosed in vivo. I think this paper illustrates that challenge by the rather limited in vivo data set presented.

1. A need to be more precise with terminology. The 4D9 antibody as I understand it is a rat monoclonal. However the human Fc version is also referred to as simply 4D9. This should be made clear and referring to the human Fc version, which is used for many studies as 4D9 makes the paper confusing.
2. In the text I do not ever see the isotype of the 4D9 rat antibody described. I might have missed it but this needs to be more clear.
3. Is the DAGDLWVPE epitope conserved between mouse and humans is it at all homologous in other TREM proteins? If there is some homology then cross-reactivity of 4D9 (with other TREM family members should be tested. Binding to the human sequence should be tested as well.
4. A somewhat picky but important concern is that the epitope mapping study does not necessarily precisely define the epitope. Binding of the other peptides to the plate is not assessed. So an easy way around this is to biotinylate the presumptive epitope and evaluate competition by the other peptides for binding to the plate bound antibody.
5. Capping or inhibition of shedding or both? Conceptually it is not clear if the effects on TREM2 function and downstream effects on microglial cells are due to increased cell surface receptor or simply sapping of the receptor. A single domain version could distinguish these mechanisms. If is mediating these effects by capping one would worry about downregulation of this response over time, which has important implications
6. Perhaps the most critical weakness is that the ex vivo lipidomic profiling from the antibody dosed 5X-mice is not adjusted for multiple testing. Only a few lipids are actually on raw p-value statistically altered, but I am almost certain most of these differences would disappear if corrected for false discovery. This data really needs to be repeated with much more sample size to know how valid it is.....
7. Other measures for target engagement in vivo are not described. One would think that such data is relevant. Indeed, cell-surface levels of TREM2, levels of sTREM2, psyk etc...seem like logical endpoints to assess.
8. Any evidence for anti-human response to the humanized 4D9?

Referee #2 (Remarks for Author):

This manuscript reports the effects of an antibody directed to the extracellular juxtamembrane region of TREM2 and prevents its cleavage from the membrane by ADAM10/17. The study is of particular interest as it provides proof of principal studies supporting the potential clinical utility of such an approach.

It should be noted that the reduction of TREM2 shedding by the antibody is only about 50-60% (Figs 1C and 3C) and this is not explicitly pointed out in the text.

In the only *in vivo* experiment the authors inject 4DP into the 5xFAD mice. The authors then evaluate its effects only on lipidomics and do not readout any of the conventional measures of TREM2 action in mice. The lipidomics data are interesting but are inadequate to allow evaluation of antibody actions on AD pathology. The authors must readout AD-related outcomes that would connect to the *in vitro* experiments performed in the rest of the manuscript and the broadly accepted actions of TREM2 in the literature. The lipidomics seem to appear out of nowhere, and the authors overinterpret these results in relation to microglial biology. The lipidomics data should include quantitation of SM, PG and other species to allow evaluation of the generality of the effect or whether its effects are dominantly on cholesterol esters. This is the most important experiment in the manuscript and the *in vivo* effects of the antibody need to be adequately documented.

One of the principal difficulties with the study is the failure to clearly differentiate the effects of elevated membrane levels of TREM2 owing to suppression of its cleavage from those arising from an agonist effect of the antibody on TREM2 signaling. This is an important distinction and could be easily evaluated in *in vitro* studies.

Specific Comments

The data in figure 1B are problematic. The western blot demonstrates a robust effect of the ADAM inhibitor and antibody treatment on the membrane levels of TREM2 compared to the DMSO control. However, treatment with GM does not have a significant effect on membrane-bound TREM2 relative to the DMSO control in the ELISA. Moreover, there are discrepancies in the effect of the isotype control and some of the antibodies observed by western analysis compared to the ELISA data. In fact, the 4D9 antibody seems to have little effect on TREM2 membrane levels compared to the Isotype Control by western blot analysis. This data set is uninterpretable.

It is hard to reconcile the abundance of the mature (glycosylated) forms of TREM2 in HEK cells (Fig 1B) and that observed in the membrane of BMDM (Fig 3B). The sTREM2 detected in the medium of the BMDM is heterogeneous with respect to molecular weight suggesting that the mature forms are abundantly expressed and shed (Fig 3C), which is inconsistent with the absence of the glycosylated TREM2 species on the cellular membranes. It seems implausible that the membrane-associated TREM2 comprises a single mature species and that antibody treatment does not result in the detection of these species on the membrane. This experiment needs to be reexamined. Also, inhibition of ADAM by GM induces an increase of both immature and mature forms of TREM2 in the membrane, however, 4D9 only seems to affect the mature form compared to isotype control (Figures 1 and 3). If 4D9 acts to inhibit ADAM cleavage of TREM2 - why is this effect on immature/mature forms so strikingly different?

The 4D9 antibody is reported to stimulate BMDM survival upon reducing CSF1 levels in culture (Fig 3D). As I understand the assay, 4D9 is bound to the plate and BMDM cultured in the absence or presence of CSF1. What is the efficiency of TREM2 engagement and signaling in this experimental context whereby an immobilized antibody has to engage the juxtamembrane region of the membrane-bound receptor? It is curious that they used ATP levels as a readout. The ATP level is reflective of metabolic activity, not cell number. Moreover, the antibody is argued to drive signaling which might predictably alter metabolism- and is not linked to cellular survival. This assay has to be performed with conventional analysis of cell number before concluding it affects cellular survival.

In the text, to address the fact that soluble TREM2 has been shown to improve microglial phenotype in an AD model, the authors state "These seemingly contradictory findings could be explained by the observation that TREM2 tends to bind to itself". However, the paper from Zhong et al. clearly indicates that this statement is not correct, sTREM2 induces the same effects in TREM2 KO and WT microglia.

I did not find the phagocytosis experiments in Figure 4 compelling. The plating densities are quite different. What was the aggregation state of Ab42? The amount of internalized Ab42 appears to be quite modest. It is not clear how the data were normalized or why Ab42 was chosen as the benchmark. The axis legend should not include the word 'increase'

Minor points:

Neither the methods or figure legend describe how the experiment in Figure 2C was conducted. Presumably, the liposomes are thought to act as ligands for TREM2, but p-Syk levels of untreated TREM2/DAP12 were not reported, only empty vector. Were the liposomes necessary to see the p-Syk signal in the 4D9-treated cells? This experiment needs a more explicit explanation.

The sorting of microglia in Fig 5 gates CD45 low cells which will capture parenchymal microglia. The plaque associated microglia that exhibit the DAM phenotype are CD45^{hi}. It is the latter cells that exhibit the most robust perturbation of metabolism and might be expected to have more severe perturbation of lipid metabolism.

Referee #3 (Remarks for Author):

In this manuscript, Schlepckow et al., develop an antibody (4D9) against the cleavage region of TREM2 with the goal of blocking the cleavage of TREM2, therefore increasing its expression on the cell membrane and decreasing soluble TREM2 levels. This approach represents a method to selectively modulate TREM2 signaling which is of great therapeutic interest for the treatment of Alzheimer's disease and other diseases of the nervous system and provides an essential and novel tool to continue exploring the role of TREM2 in these diseases. This is an exciting and timely paper that will be of broad interest of the field. However, a few key points should be addressed before publication that should be fairly straightforward to address:

1) Confirm the specificity of the antibody.

a. The authors confirm that 4D9 is specific to TREM2 (in transformed HEK cells), confirming it does not affect the expression of other transmembrane protein such as TREM1 is essential. It would be good to add additional control the to the validation of Trem2 activation via pSyk mechanism (Figure 2). Ie measure pSyk with only DAP12 introduced and not Trem2 following treatment.

Also, it is important to show the conservation between mouse and human TREM2 protein sequence in the binding region of 4D9. Protein sequence analysis could also show that other transmembrane protein, such as TREM1, are not targeted in a supplementary figure.

2) Confirm that 4D9 is blocking ADAM10/17 cleavage.

The binding sequence of 4D9 points towards an inhibition of ADAM10/17 can you confirm that 4D9 has no effect in the absence of ADAM10/17 ?

3) Phagocytosis

To fully assert that 4D9 is only affecting TREM2 dependant phagocytosis, E. coli uptake should be evaluated for the same time points as myelin and AB. Figure 4C, 4D, and 4E all need to include quantification of the "No Ab" treatment.

4) Addition how does 4D9 impact microglia response to amyloid in vivo? Ideally this or other relevant assays should be examined and presented. Also, confirming the concentration of 4D9 in the whole brain, not just in the cerebellum is important as fully dissecting choroid plexus and meningeal tissue from the cerebellum is difficult and could skew measurement of brain antibody concentration.

5) Is partial loss of expression of TREM2 be rescued by 4D9. This experiment is not essential but some sort of rescue would strengthen the paper if possible to address in a timely manner.

Minor comments

1) Statistical analysis of Fig 3A and 4A is needed.

2) Please Specify the number of animals used for each experiment.

3) Show FACS parameters used to identify the microglia population in Figure 5 and state how many cells were isolated for this experiment

Referee #1 (Comments on Novelty/Model System for Author):

see comments to authors

however technical quality gets dinged for the statistical analysis of the lipidomic data with no correction for false discovery

The lipidomic data were replaced by a significant amount of new *in vivo* data demonstrating target engagement (**new Figures 5 & 6**) and impacts on amyloidogenesis (**new Fig. 7**) in a mouse model for AD pathology (see also below).

Although there is only one other current publication of a TREM2 agonist antibody (there may be more), at least 10 other groups are doing similar things,

This argument does not consider that we are the first to present a full and comprehensive analysis of the mechanism of action of a potential TREM2 therapeutic antibody along with *in vivo* data in AD mouse models. Although we agree that it is likely that 10 or more other groups are also working on TREM2 antibodies, as none of this work has yet been published, we do not see it as relevant towards the novelty of this work. In fact, it provides an argument that this work should be published in order to bring more data on the therapeutic modulation of TREM2 into the public domain.

There is a clear need in the literature for elucidation of the mechanism of TREM2 antibodies that could provide a novel therapeutic strategy for a major unmet medical need such as AD and potentially even peripheral diseases. Specifically, after the recent release of the new Aducanumab data, demonstrating positive effects on cognition only at the highest dose, it becomes even more important to modulate microglial activity. In that regard we want to point out that we demonstrated earlier that antibody mediated Abeta clearance is TREM2 dependent (Xiang et al, EMBO Mol Med; 2016), thus a combinatorial treatment with anti-Abeta and anti-TREM2 antibodies may turn out to be the most efficacious way to treat AD. A corresponding paragraph describing this strategy has been added to the discussion.

and there is nothing really special about this antibody at this point...so again somewhat novel but not completely

We wholeheartedly disagree, there is not a single paper describing a detailed analysis of a TREM2 modulating antibody including mechanism of action, identification of the epitope, kinetics of surface binding, phospho-SYK signaling, induction of survival, inhibition of shedding, myelin and Abeta uptake, *in vivo* PK, and finally *in vivo* target engagement (see **new Figures 5 & 6**) and effects on amyloidogenesis (**new Fig. 7**). This has not been shown in publications or patents. The paper by Cheng et al. (JBC, 2018) indeed describes similar effects on survival and syk-signaling, which we clearly acknowledged in our paper. However, neither the binding site nor the mechanisms of action of this antibody are known and absolutely no *in vivo* functional or target engagement data were included in this study.

Without definitive data for target engagement in vivo the medical impact is unclear.

We have addressed this point in multiple way through the addition of new *in vivo* data. First, we have determined the ratio of 4D9 bound sTREM2 versus total sTREM2 in mouse CSF. This represents a target engagement assay that could potentially be translated to clinical studies. We show almost complete binding of 4D9 to its target *in vivo*. These data are now shown in the **new Fig. 5C-E** and described in an additional chapter added to the end of the results section. Moreover, we also include new data in the revised manuscript demonstrating that 4D9 treatment leads to an increase of total TREM2 in brain, further confirming the impact of this antibody on Trem2 biology *in vivo* (**new Fig. 5F**). We further generated evidence of a pharmacodynamic response resulting from TREM2 target engagement in the CNS after treatment of an APP knock-in mouse model with antibody 4D9 by immunohistochemically demonstrating increased microglial TREM2 and decreased P2RY12 expression, two markers for a disease associated and homeostatic mRNA signatures (**new Fig. 6**).

Finally, we now demonstrate that 4D9 reduces amyloidogenesis in a mouse model for AD pathology (new Fig. 7).

The model system issue is more of general issue for the AD field, how would we determine if the antibody is beneficial?

Altering some microglial phenotype does not mean the efficacy would be apparent in AD.

This is a difficult point for all therapeutics tested presently and in the past. We believe that the only way to definitively prove a therapeutic effect on AD is a clinical trial. However, that needs first careful preclinical evidence for a clear mechanism of action, pharmacokinetics that demonstrably connect to a pharmacodynamic response, and evidence for *in vivo* target engagement including amyloidogenesis. That's exactly what we provide now in the new Figures 5, 6 and 7.

Referee #1 (Remarks for Author):

This paper describes the identification of a TREM2 binding antibody that appears to inhibit shedding of mouse TREM2 and increase the cell surface levels of TREM2. Convincing data is provided that suggest that the antibody shows high affinity and epitope mapping is performed. Short term PK-PD studies are performed as well as in vitro/ex vivo evidence for functional activity following binding to either transfected cells or primary microglial cells. In vivo studies claim to show alterations in abnormal lipid profiles in 5X FAD mice, but these data are problematic due to lack of correction for multiple testing.

We agree that the lipid data as presented in the original version of the manuscript could be challenging to interpret. In accordance with our revision plan and the recommendations of the editor, we removed the lipid data and now focus the entire manuscript on the characterization of the antibody, its mechanism and cellular modulating activities, pharmacokinetics and demonstration of *in vivo* target engagement through multiple approaches (see new Figures 5 & 6). Our revision also includes the demonstration of the ability of antibody 4D9 to reduce amyloid plaque formation in a mouse model for AD pathology (see new Fig. 7).

Based on the hypothesis that partial loss of function of TREM2 underlies Alzheimer's risk associated with TREM2 AD risk variants, a number of groups are isolating potentially TREM2 agonist antibodies. Indeed, a recent report (Cheng JBC 2018 describes at least one). Clearly, a challenge for the field is what are the readouts for successful target engagement of such antibodies when dosed in vivo. I think this paper illustrates that challenge by the rather limited in vivo data set presented.

We have addressed this point by determining the ratio of 4D9 bound sTREM2 versus total sTREM2 in mouse CSF. We show almost complete binding of 4D9 to its target *in vivo*. These data are now shown in the new Fig. 5C-E and described in an additional paragraph added to the end of the results section. Moreover, we show additionally that 4D9 treatment leads to a dose dependent increase of total TREM2 in brain (new Fig. 5F). We further confirmed target engagement after treatment of an APP knock-in mouse model with antibody 4D9 by immunohistochemically demonstrating increased microglial TREM2 and decreased P2RY12 expression, two markers for a disease associated and homeostatic mRNA signatures (new Fig. 6). Furthermore, in the new Fig. 7 we now show that antibody 4D9 reduces amyloidogenesis in the APP knock-in mouse model.

1. A need to be more precise with terminology. The 4D9 antibody as I understand it is a rat monoclonal. However, the human FC version is also referred to as simply 4D9. This should be made clear and referring to the human Fc version, which is used for many studies as 4D9 makes the paper confusing.

This has been corrected in the revised version.

2. In the text I do not ever see the isotype of the 4D9 rat antibody described. I might have missed it but this needs to be more clear.

We now described the isotype of antibody 4D9 in the results section of the manuscript. Please also note that a table is shown within the Materials and Methods indicating the isotype of all antibodies described.

3. Is the DAGDLWVPE epitope conserved between mouse and humans is it at all homologous in other TREM proteins? IF there is some homology then cross-reactivity of 4D9 (with other TREM family members should be tested. Binding to the human sequence should be tested as well.

We have added new data (**new Fig. 1F, upper panel**), which show significant sequence conservation between mouse and human TREM2 around the epitope of 4D9. However, we found a lack of species or family member cross reactivity of 4D9 to human TREM2 and mouse TREM1 by western blotting and BIAcore (**new Fig. 1F, lower panel and new Fig 1H**). Thus, 4D9 specifically detects mouse TREM2.

4. A somewhat picky but important concern is that the epitope mapping study does not necessarily precisely define the epitope. Binding of the other peptides to the plate is not assessed. So an easy way around this is to biotinylate the presumptive epitope and evaluate completion by the other peptides for binding to the plate bound antibody.

This has been addressed by a peptide competition assay shown in the **new Fig. 1G**.

5. Capping or inhibition of shedding or both? Conceptually it is not clear if the effects on TREM2 function and downstream effects on microglial cells are due to increased cell surface receptor or simply sapping of the receptor. A single domain version could distinguish these mechanisms. If is mediating these effects by capping one would worry about downregulation of this response over time, which has important implications

We added data using the Fab fragments of antibody 4D9 to the completely **new Fig. 2A-F**. Our findings indicate that the monovalent 4D9 Fab fragment binds to cell surface TREM2 but is not capable of inducing pSYK signaling and also fails to inhibit TREM2 shedding in a cell-based assay. However, the Fab fragment can block in vitro TREM2 cleavage by ADAM17. Therefore, we now demonstrate a key mechanistic insight for 4D9 function which requires bivalent TREM2 binding. We suggest 4D9 acts via dual mechanisms for boosting TREM2 dependent activities: (1) cross-linking of TREM2 on the plasma membrane which drives receptor activation, and (2) inhibition of ADAM 10/17 cleavage either by steric hindrance and/or by TREM2 dimerization.

6. Perhaps the most critical weakness is that the ex vivo lipidomic profiling from the antibody dosed 5X-mice is not adjusted for multiple testing. Only a few lipids are actually on raw p-value statistically altered, but I am almost certain most of these differences would disappear if corrected for false discovery. This data really needs to be repeated with much more sample size to know how valid it is.....

As mentioned above, in accordance with our revision plan and the recommendations of the editor, we removed the lipid data and now focus the entire manuscript on the characterization of the antibody MOA, its modulating activities, pharmacokinetics and *in vivo* target engagement including effects on amyloidogenesis (see **new Figures 5, 6 and 7**).

7. Other measures for target engagement in vivo are not described. One would think that such data is relevant. Indeed, cell-surface levels of TREM2, levels of sTREM2, psyk etc...seem like logical endpoints to assess.

As mentioned above, we addressed this point by determining the ratio of 4D9 bound sTREM2 versus total sTREM2 in mouse CSF. We show almost complete binding of 4D9 to its target *in vivo*. These data are now shown in the **new Fig. 5C-E** and described in an additional paragraph added to the end of the results section. Moreover, we also include new data in the revised manuscript demonstrating that 4D9 treatment leads to a dose dependent increase of total TREM2 in brain, further confirming the impact of this antibody on Trem2 biology in vivo (**new Fig. 5F**).

We provide further evidence of the impact of TREM2 target engagement in the CNS after treatment of an APP knock-in mouse model with antibody 4D9 by immunohistochemically demonstrating

increased microglial TREM2 and decreased P2RY12 expression, two markers for a disease associated and homeostatic mRNA signatures, as well as decreased amyloidogenesis (**new Fig. 6 & new Fig. 7**).

8. Any evidence for anti-human response to the humanized 4D9?

At acute dosing time courses, we have not observed any anti-drug response and have specifically noted this in the revised manuscript. Typical ADA responses lead to increased clearance of the antibody in plasma which was not observed in a PK study with 4D9 under the dosing durations we report.

Referee #2 (Remarks for Author):

This manuscript reports the effects of an antibody directed to the extracellular juxtamembrane region of TREM2 and prevents its cleavage from the membrane by ADAM10/17. The study is of particular interest as it provides proof of principal studies supporting the potential clinical utility of such an approach.

It should be noted that the reduction of TREM2 shedding by the antibody is only about 50-60% (Figs 1C and 3C) and this is not explicitly pointed out in the text.

This is now pointed out in the Results.

In the only in vivo experiment the authors inject 4DP into the 5xFAD mice. The authors then evaluate its effects only on lipidomics and do not readout any of the conventional measures of TREM2 action in mice. The lipidomics data are interesting but are inadequate to allow evaluation of antibody actions on AD pathology. The authors must readout AD-related outcomes that would connect to the in vitro experiments performed in the rest of the manuscript and the broadly accepted actions of TREM2 in the literature. The lipidomics seem to appear out of nowhere, and the authors overinterpret these results in relation to microglial biology. The lipidomics data should include quantitation of SM, PG and other species to allow evaluation of the generality of the effect or whether it effects are dominantly on cholesterol esters. This is the most important experiment in the manuscript and the in vivo effects of the antibody need to be adequately documented.

As addressed in the points raised by reviewer 1, and in accordance with our revision plan and the recommendations of the editor, we removed the lipid data and now focus the entire manuscript on the characterization of the antibody, its modulating activities, pharmacokinetics and *in vivo* target engagement (see **completely new Figures 5,6 & 7**). This new data includes the demonstration of the ability of antibody 4D9 to reduce amyloid plaque formation in a mouse model for AD pathology (see **new Figure 7**). Moreover, in the same mice we also observed that 4D9 increased TREM2 and decreased P2RY12 expression, suggesting that the antibody boosts a protective DAM signature (see **new Fig. 6**).

One of the principal difficulties with the study is the failure to clearly differentiate the effects of elevated membrane levels of TREM2 owing to suppression of its cleavage from those arising from an agonist effect of the antibody on TREM2 signaling. This is an important distinction and could be easily evaluated in in vitro studies.

As described above (reviewer 1), we have generated new data using 4D9 derived Fab fragments to address this point. These data have been added to the **new Figures 2A-F**. Our findings indicate that the monovalent 4D9 Fab fragment binds to cell surface TREM2 but does not induce pSyk signaling and also fails to inhibit TREM2 shedding. However, since the antibody blocks *in vitro* TREM2 cleavage by ADAM17, we suggest a dual mechanism for boosting TREM2 dependent activities: (1) cross-linking of TREM2 on the plasma membrane, and (2) inhibition of ADAM 10/17 cleavage either by steric hindrance and/or by TREM2 dimerization.

Specific Comments

The data in figure 1B are problematic. The western blot demonstrates a robust effect of the ADAM inhibitor and antibody treatment on the membrane levels of TREM2 compared to the DMSO control. However, treatment with GM does not have a significant effect on membrane-bound TREM2 relative to the DMSO control in the ELISA. Moreover, there are discrepancies in the effect of the isotype control and some of the antibodies observed by western analysis compared to the ELISA data. In fact, the 4D9 antibody seems to have little effect on TREM2 membrane levels compared to the Isotype Control by western blot analysis. This data set is uninterpretable.

In the new Fig. 1B and C, we now carefully quantitated the effects of antibody 4D9 on sTREM2 and membrane bound TREM2. In addition, multiple samples from GM, isotype and 4D9 treatments are shown by western blotting in the upper panel for both, effects on membrane bound TREM2 and sTREM2.

It is hard to reconcile the abundance of the mature (glycosylated) forms of TREM2 in HEK cells (Fig 1B) and that observed in the membrane of BMDM (Fig 3B).

As shown in our previous publications (Kleinberger et al., Sci Trsl Med, 2014; Kleinberger et al. EMBO J 2017), the glycosylation pattern of TREM2 in different cell types is often quite divergent. Therefore, the pattern of HEK and BMDM derived glycosylated TREM2 cannot be directly compared. However, upon deglycosylation immature and mature TREM2 run as single unique species in both, HEK293 and BMDMs as shown below.

The sTREM2 detected in the medium of the BMDM is heterogenous with respect to molecular weight suggesting that the mature forms are abundantly expressed and shed (Fig 3C)

What varies are the bands from the light chains of the two antibodies, not sTREM2. This is also seen in the new Fig. 1C. We now point this out in the description of the data in the figure legends to Fig. 1C and Fig. 3C to avoid confusion.

, which is inconsistent with the absence of the glycosylated TREM2 species on the cellular membranes.

There is in fact mature TREM2 present on the membrane fractions, this is seen with the smear above the mature band specifically after 4D9 treatment. Surface TREM2 is rapidly cleaved by ADAM10, which makes the detection of the full glycosylated TREM2 version difficult. We added a longer exposure to visualize fully glycosylated TREM2 that appears as a smear like in HEK293 cells (new Fig. 3B).

It seems implausible that the membrane associated TREM2 comprises a single mature species and that antibody treatment does not result in the detection of these species on the membrane. This experiment needs to be reexamined.

We disagree with this point, as we do see an increase of fully glycosylated mature TREM2 in Fig. 3B. As noted above, we have shown a longer exposure of the immunoblot in the revised manuscript.

Also, inhibition of ADAM by GM induces an increase of both immature and mature forms of TREM2 in the membrane, however, 4D9 only seems to affect the mature form compared to isotype control (Figures 1 and 3). If 4D9 acts to inhibit ADAM cleavage of TREM2 - why is this effect on immature/mature forms so strikingly different?

In Fig. 3 we did not use GM treatment. In the **new Fig. 1** we do not see any difference between 4D9 and GM treatment. GM and 4D9 mostly affect mature TREM2, which is consistent with the fact that ADAM10/17 cleaves predominantly on the cell surface.

The 4D9 antibody is reported to stimulate BMDM survival upon reducing CSF1 levels in culture (Fig 3D). As I understand the assay, 4D9 is bound to the plate and BMDM cultured in the absence or presence of CSF1. What is the efficiency of TREM2 engagement and signaling in this experimental context whereby an immobilized antibody has to engage the juxtamembrane region of the membrane bound receptor?

It is very difficult to measure the TREM2 surface engagement of antibody and receptor in these in vitro plate coated studies, and therefore provides only an approximate potency value for these antibodies. We noted this more clearly in the revised text. The p-Syk assay that utilizes soluble antibody provides a more accurate measure of functional potency. Therefore, we have used the latter assay to guide antibody concentrations.

We have made many attempts to optimize the survival assay to a solution-based system where we could directly compare to pSYK or FACS based assessments of target engagement. However, at present only formats where antibody is plate coated induce a survival phenotype. This suggests that significant receptor clustering is required to achieve a signaling threshold that translates functionally to survival of BMDM. Although we do not fully understand how the TREM2 clustering elicited in this setup translates in vivo, we believe this is a highly valuable assay for determining effects on TREM2 function and has been used by a number of groups to identify/validate TREM2 ligands (e.g. Zhao et al., Neuron 2018). Importantly, the findings from this assay are consistent with the functional data observed in the pSYK and phagocytosis assays and therefore provides important contributions to the data package supporting Trem2 agonist activity via 4D9 in primary cells.

One additional point of clarification, the survival assay is run in low doses of MCSF where a survival defect has been demonstrated with TREM2 KO BMDM, demonstrating the TREM2 dependence of this phenotype.

It is curious that they used ATP levels as a readout. The ATP level is reflective of metabolic activity, not cell number. Moreover, the antibody is argued to drive signaling which might predictably alter metabolism- and is not linked to cellular survival. This assay has to be performed with conventional analysis of cell number before concluding it affects cellular survival.

This is a conventional assay to measure cell viability.

“The CellTiter-Glo® Assay generates a "glow-type" luminescent signal, which has a half-life generally greater than five hours, depending on cell type and medium used. The extended half-life eliminates the need to use reagent injectors and provides flexibility for continuous or batch mode processing of multiple plates. The unique homogeneous format avoids errors that may be introduced by other ATP measurement methods that require multiple steps.”

Nevertheless, to avoid any potential confounding effects of TREM2 activity on ATP levels in microglia we also included images and associated quantification of isotype and 4D9 treated cells in the figure shown below to demonstrate that 4D9 indeed impacts viability rather than ATP production.

In the text, to address the fact that soluble TREM2 has been shown to improve microglial phenotype in an AD model, the authors state "These seemingly contradictory findings could be explained by the observation that TREM2 tends to bind to itself". However, the paper from Zhong et al. clearly indicates that this statement is not correct, sTREM2 induces the same effects in TREM2 KO and WT microglia.

We apologize for the confusion, as the 2017 but not the 2019 paper from this group showed the knockout. We corrected this statement in the revised manuscript. However, we also would like to note that sTREM2 may not be as protective. The TREM2 H157Y variant, which is associated with an enhanced risk for AD, increases shedding of TREM2 (Schlepckow et al., EMBO Mol Med 2017). This suggests that increased sTREM2 is not sufficient for mitigating AD risk but instead exacerbates it.

I did not find the phagocytosis experiments in Figure 4 compelling. The plating densities are quite different. What was the aggregation state of Ab42? The amount of internalized Ab42 appears to be quite modest. It is not clear how the data were normalized or why Ab42 was chosen as the benchmark. The axis legend should not include the word 'increase'

We want to point out that the quantitation in Figs. 4C-E shows the percentage increase in the number of substrate positive cells as compared to the treated cells in the absence of antibody. Thus, the data are properly normalized and differences in cell numbers, which may be caused by 4D9's promotion of cell survival, do not interfere with the quantitation.

We have now added nuclear staining to better show the number of cells in each condition (**new Fig. 4B**).

We used Abeta42, which after overnight incubation at 37°C, 800rpm was aggregated to oligomers and fiber-like structures. We followed the exact same protocol as used in our previous paper (Kleinberger et al. 2014), where we demonstrated that disease-associated TREM2 variants significantly lower cellular uptake of aggregated Abeta42.

Minor points:

Neither the methods or figure legend describe how the experiment in Figure 2C was conducted. Presumably, the liposomes are thought to act as ligands for TREM2, but p-Syk levels of untreated TREM2/DAP12 were not reported, only empty vector. Were the liposomes necessary to see the p-Syk signal in the 4D9-treated cells? This experiment needs a more explicit explanation.

Liposomes were not necessary to see p-Syk signaling as shown in the **new Fig. 2D** (former Fig. 2A). Liposomes rather act as TREM2 signal activating ligands. We also added a reference, which describes liposome mediated effects on p-Syk signaling (Shirotani et al., Sci Rep 2019) and include a detailed methodological description in Materials and Methods.

The sorting of microglia in Fig 5 gates CD45 low cells which will capture parenchymal microglia.

The plaque associated microglia that exhibit the DAM phenotype are CD45hi. It is the latter cells that exhibit the most robust perturbation of metabolism and might be expected to have more severe perturbation of lipid metabolism.

In accordance with our revision plan and the recommendations of the editor, we removed the lipid data (see above).

Referee #3 (Remarks for Author):

In this manuscript, Schlepckow et al., develop an antibody (4D9) against the cleavage region of TREM2 with the goal of blocking the cleavage of TREM2, therefore increasing its expression on the cell membrane and decreasing soluble TREM2 levels. This approach represents a method to selectively modulate TREM2 signaling which is of great therapeutic interest for the treatment of Alzheimer's disease and other diseases of the nervous system and provides an essential and novel tool to continue exploring the role of TREM2 in these diseases. This is an exciting and timely paper that will be of broad interest of the field. However, a few key points should be addressed before publication that should be fairly straightforward to address:

1) Confirm the specificity of the antibody.

a. The authors confirm that 4D9 is specific to TREM2 (in transformed HEK cells), confirming it does not affect the expression of other transmembrane protein such as TREM1 is essential.

We have added the requested data in the **new Fig. 1F** and the **new Fig. 1H** and show that 4D9 does not cross-react with human TREM2 or mouse TREM1 (see also point #3 by reviewer 1).

It would be good to add additional control the to the validation of Trem2 activation via pSyk mechanism (Figure 2). Ie measure pSyk with only DAPI2 introduced and not Trem2 following treatment.

DAPI2 alone will not reach the cell surface due to the absence of its co-receptor TREM2 (Lanier et al, *Immunity* 1998), which upon binding neutralizes the negative charge within the transmembrane domain of DAPI2. Thus, this experiment is not likely to be informative. Figure 2E contains all controls required to rule out a non-specific effect including empty vector, isotype control and no antibody.

Also, it is important to show the conservation between mouse and human TREM2 protein sequence in the binding region of 4D9. Protein sequence analysis could also show that other transmembrane protein, such as TREM1, are not targeted in a supplementary figure.

We show in the **new Fig. 1F** that although the 4D9 epitope is conserved between mouse and human TREM2 (upper panel), antibody 4D9 does not detect human TREM2 (lower panel). Antibody 4D9 also does not detect mouse TREM1 (see **new Fig. 1F**). Western blot data demonstrating lack of cross reactivity has also been confirmed by BIAcore (**new Fig. 1H**).

2) Confirm that 4D9 is blocking ADAM10/17 cleavage.

We conducted in vitro cleavage assays using TREM2 peptides covering the cleavage site and recombinant ADAM17. In the **new Fig. 2B** we now demonstrate that 4D9 inhibits in vitro cleavage of TREM2 by ADAM17.

The binding sequence of 4D9 points towards an inhibition of ADAM10/17 can you confirm that 4D9 has no effect in the absence of ADAM10/17?

Our new experiments with the Fab fragments in the **revised Fig. 2A-F** indicate that TREM2 cross-linking by 4D9 inhibits access to the sheddase but also increases TREM2 signaling. Therefore, the antibody is likely to act via a combination of mechanisms and would thus still affect TREM2 function via bivalent receptor binding and clustering in the absence of ADAM10/17.

3) Phagocytosis

To fully assert that 4D9 is only affecting TREM2 dependent phagocytosis, E. coli uptake should be evaluated for the same time points as myelin and Abeta. Figure 4C, 4D, and 4E all need to include quantification of the "No Ab" treatment.

The "No Ab" treatment was already quantified and was taken as baseline for the isotype and 4D9 quantifications (see Materials and Methods). The different time points were chosen since uptake of the different substrates will involve different receptors and different pathways and therefore occur at different rates. The different size of each substrate will affect the uptake rate additionally. Uptake of E.coli pHrodo occurs more rapidly than myelin or Abeta.

4) Addition how does 4D9 impact microglia response to amyloid in vivo? Ideally this or other relevant assays should be examined and presented.

We have addressed this issue by treating APP knockin mice with antibody 4D9 (see **new Figures 6 and 7**). We now demonstrate that antibody 4D9 reduces amyloid plaque formation in a mouse model for AD pathology (see **new 7**). Moreover, in the same mice we also observed that 4D9 increased TREM2 and decreased P2RY12 expression, suggesting that the antibody boosts a protective DAM signature (see **new Fig. 6**).

Also, confirming the concentration of 4D9 in the whole brain, not just in the cerebellum is important as fully dissecting choroid plexus and meningeal tissue from the cerebellum is difficult and could skew measurement of brain antibody concentration.

We now determined the concentration of 4D9 in the entire brain as requested (see **new Fig. 5B**).

5) Is partial loss of expression of TREM2 be rescued by 4D9. This experiment is not essential but some sort of rescue would strengthen the paper if possible to address in a timely manner.

This has been addressed in the **new Fig. 3D**, where we now show that 4D9 can also increase survival of heterozygous TREM2 knockout BMDMs.

Minor comments

1) Statistical analysis of Fig 3A and 4A is needed.

Selective binding of antibody 4D9 on the cell surface of BMDMs was quantitated as requested. The new results including statistical analysis are shown in the **new Fig.3A**.

We are not making a statistical argument with the data in 4A, lack of binding in the TREM2 knockout confirms the specificity of binding to wild type primary microglia.

2) Please Specify the number of animals used for each experiment.

Numbers of animals used for each experiment are now included.

3) Show FACS parameters used to identify the microglia population in Figure 5 and state how many cells were isolated for this experiment

These experiments were removed in accordance with the revision plan.

Thank you for the submission of your revised manuscript to EMBO Molecular Medicine. We have now received the enclosed reports from the referees that were asked to re-assess it. As you will see the reviewers are now globally supportive and I am pleased to inform you that we will be able to accept your manuscript pending final amendments.

1) Please address referee #1's comments in writing. After consulting with referee #2 about referee #1's remaining comments, we agreed to only ask you to rewrite the main conclusions and discussion to reflect referee #1's concerns. Indeed, referee #2 recognizes the validity of referee #1 concerns, while still appreciating the effect of the antibody, with a good number of repetitions. More could be done indeed, like including Iba1 staining and analyzing Abeta internalization; should you have this data already, we would encourage you to add it to the paper. Otherwise, please include a written statement in the main body of the article.

Please provide a point-by-point letter INCLUDING my comments as well as the reviewer's reports and your detailed responses to their comments (as Word file).

***** Reviewer's comments *****

Referee #1 (Comments on Novelty/Model System for Author):

my only issue is that the data on the APP mouse study is very superficial

Referee #1 (Remarks for Author):

The first version of the paper had many data gaps in it and lacked convincing evidence for target engagement in the brain, the authors have done an excellent job of responding to the critiques. Critical data not in the first manuscript is now included and the data is for the most part quite convincing.

There still remain some concerns that are addressable. These concern the lack of consideration of sex as a variable, even in post hoc analysis.

I am also only partially convinced by the APP mouse study. The documentation that the 4D9 antibody is causing "plaque" clearance in 10 days: again sex is not assessed as a variable here, and the documentation that diffuse deposits around the more cored plaques are cleared is really not clear from the images shown.

There is no biochemical analysis, no co-stain of microglia and the plaques. One asks where did the Abeta go, if it is being phagocytosed, is it detectable in microglial cells as others have shown?

Standard biochemistry for different soluble Abeta pools and APP processing is missing.

If these effects are this robust in 10 days then these studies are easy to repeat.

At the end of the day most in the field will focus on this final data item, it should be completely convincing and robust. It is a quite remarkable observation if replicable and documented better

Referee #2 (Remarks for Author):

The revised manuscript bears little resemblance to the original. The authors have done an excellent job in responding to the critiques. I have no criticisms.

2nd Revision - authors' response

12th Feb 2020

Referee #1

lack of consideration of sex as a variable, even in post hoc analysis.

We re-analyzed our data considering sex as a variable (Figs 3, 6, and 7) but did not find any sex-specific differences. In Fig. 5 we only used male mice. The corresponding information was given in the figure legends.

documentation that diffuse deposits around the more cored plaques are cleared is really not clear from the images shown. There is no biochemical analysis, no co-stain of microglia and the plaques.

One asks where did the Abeta go, if it is being phagocytosed, is it detectable in microglial cells as others have shown? Standard biochemistry for different soluble Abeta pools and APP processing is missing.

We have addressed these points as follows:

- (1.) We added the **new Fig. 7C** with a higher magnification of the brain sections to the paper, which now shows clearance of the plaque halo (but not the plaque core) much better. We note that the background of the staining is indeed very low. The contrast of the images added in the manuscript has not been changed. Decreasing the contrast or increasing the brightness of the images yields a false perception of higher amounts of plaques.
- (2.) We also performed the requested biochemical experiments to confirm the reduction of soluble Abeta in the **new Fig. 7 F-H**. In Fig. 7F we now show a western blot of soluble (DEA-extracted) versus insoluble (formic acid extracted) Abeta. The western blot confirms selective clearance of soluble Abeta. In the **new Figs 7G and 7H** we quantitated the corresponding findings.

We can currently not address the fate of engulfed Abeta. We rather believe that this should be part of a new manuscript. Nevertheless, we refer to this critical point in the Discussion.

If these effects are this robust in 10 days then these studies are easy to repeat.

We would like to note that according to German animal laws, a repetition of these experiments requires a new application for a license. This takes at least half a year. Only then we can start to setup new breedings. Therefore, a repetition would take at least one year.

Referee #2

No more concern raised.

The authors performed the requested editorial changes.

Corresponding Author Name: Christian Haass, Joseph W. Lewcock

Manuscript Number: EMM-2019-11227-V2